# Cluster Analysis of Vertical Polarimetric Radio Occultation Profiles and Corresponding Liquid and Ice Water Paths From GPM Microwave Data

Jonas E. Katona[1,2], Manuel de la Torre Juárez[2], Terence L. Kubar[2,3], F. Joseph Turk[2], Kuo-Nung Wang[2], and Ramon Padullés[4]

[1]Yale University, Applied Mathematics Program
[2]NASA Jet Propulsion Laboratory, California Institute of Technology
[3]University of California, Los Angeles, Joint Institute for Regional Earth Systems Science and Engineering
[4]Institut de Ciencies de l'Espai, Consejo Superior de Investigaciones Cientificas

**Correspondence:** Jonas E. Katona (jonas.katona@yale.edu) and Manuel de la Torre Juárez (mtj@jpl.nasa.gov)

**Abstract.** Polarimetric Radio Occultations (PRO) of the Global Navigation Satellite System are able to characterize precipitation structure and intensity. Prior studies have shown the relationship between precipitation and water vapor pressure columns, known as the "precipitation pickup." Less is known about the relationship between the vertical distributions of temperature and moisture globally within precipitating scenes as measured from space. This work uses cluster analysis of PRO to explore how the vertical distributions of temperature and moisture—combined into PRO refractivity—relate to vertical distributions of precipitation and moisture variables. We evaluate the ability of $k$-means clustering to find relationships among PRO polarimetric phase difference, refractivity, liquid water path (LWP), ice water path (IWP), and water vapor pressure using over two years of data matched between the Global Precipitation Measurement (GPM) mission from the Radio Occultations (RO) and Heavy Precipitation demonstration mission onboard the Spanish Paz spacecraft (ROHP-PAZ). A polytropic potential refractivity model for polytropic atmospheres is introduced to ascertain how different vertical thermodynamic profiles that can occur during different precipitation scenarios are related to changes in the polytropic index and thereby vertical heat transfer rates. The cluster analyses suggest a relationship between the amplitude and shape of deviations from the potential refractivity model and water vapor pressure. These analyses also confirm a positive correlation between vertical shapes of polarimetric phase difference and both LWP and IWP. For certain values, the coefficients of the polytropic potential refractivity model flag physical vs. non-physical retrievals, and indicate when a profile has little to no moisture. The study reveals a similar relationship between the clustering for these coefficients and different water vapor pressure profiles.

## 1 Introduction

General circulation models need to represent the spatiotemporal structure of precipitation for accurate predictions of climate variability and deep convective structures. Models and observations show a relationship in the probability densities between the precipitation and column water vapor relationship known as the precipitation pickup. Emmenegger et al. (2022) conclude that the majority of the models' convection onset statistics display some degree of temperature dependence in the column

water vapor value of the pickup and collapse approximately to a common critical column relative humidity value across saturation mixing ratio bins. However, prior results suggest that the onset of convective instability has a complex dependence on temperature. The vertical structure of temperature and moisture, as well as the entrainment of free tropospheric air, affect the buoyancy of a rising convective plume, yielding an onset moisture-temperature dependence slightly different than that of bulk saturation. This work explores how the vertical structure of temperature and moisture, combined into refractivity measured by RO, relates to distributions of precipitation and moisture variables.

Global Navigational Satellite System (GNSS) satellites orbiting Earth periodically send circularly polarized radio signals indicating their positions globally. As these satellites occult from a low Earth-orbiting satellite with a GNSS receiver, the radio signal they receive has been refracted and bent by the atmosphere. The bending angle is caused by the atmospheric refractivity gradient in the region where the signal traveled. The degree of bending can be calculated using the geometry between the emitting satellite and a receiver, and the shift in signal phase between when the signal is emitted and received. GNSS radio occultations (RO) provide refractivity, $N$, which is related to pressure ($p$, in hPa), temperature ($T$, in K), and water vapor pressure ($e$, in hPa) as follows (e.g., Smith and Weintraub, 1953; Kliore et al., 1974) for an atmospheric air composition with approximately 78 percent nitrogen and 21 percent oxygen containing water:

$$N = \frac{k_1 p}{T} + \frac{k_2 e}{T^2}, \tag{1}$$

where $k_1 = 77.6$ and $k_2 = 3.73 \times 10^5$ are typically given without dimensions. However, $N$ is expressed in refractivity units, N-units; hence, $k_1$ would be understood in N-units·K/hPa, and $k_2$ in N-units·K$^2$/hPa.

Quantities derived from RO have demonstrated high accuracy and resolution in space (e.g., Kursinski et al., 1997; Huang et al., 2010; Son et al., 2017). RO temperatures derived from refractivity have been shown to be of similar quantitative accuracy as temperatures directly measured by radiosondes, which are mostly limited to land (e.g., Nishida et al., 2000; Randel et al., 2003; Schmidt et al., 2004; Kim and Son, 2012).

One of the most powerful applications of RO has been in understanding climatic trends—including intraseasonal-to-interannual atmospheric modes of variability such as the quasi-biennial oscillation (QBO), Madden–Julian oscillation (MJO), and El Niño–Southern Oscillation (ENSO)—as they relate to atmospheric structure over the tropics (Scherllin-Pirscher et al., 2021), especially in the upper-troposphere–lower-stratosphere (UTLS) region (Schmidt et al., 2004; Lackner et al., 2011; Johnston et al., 2018, 2022). RO observations have also been used to uncover and measure the upper-level thermal structures of deep convection in tropical storms both alongside and without precipitation radar data (Biondi et al., 2012; Xian and Fu, 2015; Scherllin-Pirscher et al., 2021). In the context of precipitation events, Johnston et al. (2018, 2022) studied the impacts of deep convection and precipitation on the thermodynamic structure of the UTLS region by collocating RO temperature profiles with data from the Global Precipitation Measurement (GPM) Mission and Tropical Rainfall Measuring Mission (TRMM) in both the tropics and mid-latitudes.

Equation (1) shows that using RO refractivity data to retrieve thermodynamic variables such as temperature, pressure, and water vapor remains underconstrained. Water vapor information is extracted from refractivity by assuming that the temperature profiles from a given weather analysis—typically either the European Centre for Medium-Range Weather Forecasts (ECMWF)

or the National Centers for Environmental Prediction (NCEP)—are accurate at the location of each RO profile, even in cases where the RO and model refractivity may differ (e.g., Kursinski et al., 1997; Kuo et al., 2001). Two common methods for extracting water vapor information from RO refractivity are the 1D-Var method (Wee et al., 2022), which iteratively refines retrievals by combining RO data with background atmospheric model information through a variational data assimilation process, and the direct method (Hajj et al., 2002), which derives retrievals based on hydrostatic equilibrium and an assumed model or background temperature profile. To avoid relying on model water vapor pressure as an assumed background a priori, this study uses the direct retrieval method assuming temperature profiles provided by NCEP.

An inaccurate refractivity profile from the analysis will lead to erroneous water vapor retrievals. Because RO has a more valuable contribution to model improvement precisely in the profiles where the weather analysis and RO differ, the relationship between water vapor and refractivity has a higher error bar, particularly in the most useful profiles. Moreover, GNSS RO measurements are sensitive to variations in temperature and water vapor within clouds (Kuo et al., 2001; Huang et al., 2010), but require other observables to confirm the presence of clouds and understand their structure.

Polarimetric Radio Occultation (PRO) provides a way to expand the applications of standard RO. PRO measures the response of circularly polarized GNSS radio signals to atmospheric anisotropies like precipitating droplets and ice crystals, as these induce a phase difference between the horizontal (H) and vertical (V) components of the GNSS radio signal. The polarimetric phase difference, $\Delta\Phi$, between H and V is related to the amount of rain drops or ice crystals in the atmosphere (Tomás et al., 2018; Cardellach et al., 2019; Wang et al., 2022; Padullés et al., 2023) using $\Delta\Phi$ and has promising applications in weather model assimilation (Ruston and Healy, 2021; Wang et al., 2022; Hotta et al., 2024), climate monitoring (Cardellach et al., 2019; Gleisner et al., 2022), and atmospheric research (Turk et al., 2021; Padullés et al., 2023). Datasets from GNSS PRO contain data on refractivity and $\Delta\Phi$, both as functions of height. Unlike infrared instruments, PRO gives data even inside clouds with a higher vertical resolution than microwave (e.g. Turk et al., 2019).

Statistical correlations as a function of height between integrated water content (or water path) from CloudSat—a NASA satellite mission to survey the vertical structure of clouds and their water content via a radar that launched on April 28, 2006, and ended on December 20, 2023 (NASA, 2024)—along the RO ray path and $\Delta\Phi$ were shown to be strong (Padullés et al., 2023). There are models for how a given thermodynamic state of the atmosphere will affect a propagating RO signal and cause a $\Delta\Phi$ (e.g., Padullés et al., 2023, and references therein). However, a precise formula is missing for how a measured $\Delta\Phi$ relates to thermodynamic atmospheric states. GNSS PRO is generally insensitive to non-dipolar and to spherically symmetric particles, such as aerosols and non-precipitating cloud droplets (Padullés et al., 2023; Hotta et al., 2024). CloudSat-based water content measurements tend to be more sensitive to these smaller particles and cloud tops (Padullés & Turk, private communication). $\Delta\Phi$ at a specific height could result from precipitation of liquid water, ice water, or both—and may still be influenced by non-precipitating features, such as anisotropic ice crystals (Padullés et al., 2023). It remains an open question whether, or to what extent, differentiating between liquid and ice water precipitation—let alone non-precipitating hydrometeors—is possible using PRO data alone. Therefore, we explore if different vertical distributions of precipitation- or moisture-related variables—$\Delta\Phi$, liquid water path (LWP), ice water path (IWP), and water vapor pressure—are interrelated.

Cluster analysis is a family of methods used to group and separate populations into different groups or "clusters" within a dataset based on some measure of similarity or hierarchy. One of the most popular clustering techniques is $k$-means clustering—a flexible, established unsupervised learning method that has been used to classify and analyze the different distributions of physical variables present in climatic and atmospheric datasets (e.g. Jakob and Tselioudis, 2003; Rossow et al., 2005; Yokoi et al., 2011; Wilks, 2019; Govender and Sivakumar, 2020; Nidzgorska-Lencewicz and Czarnecka, 2020).

This study uses cluster analysis to look at how the vertical shape of $\Delta\Phi$ along the RO ray correlates with that of other thermodynamic variables such as refractivity, water vapor pressure, liquid water path (LWP), and ice water path (IWP) along the ray as functions of height at given latitudes and longitudes. A $k$-means cluster analysis is performed to see if cluster centroids relate to physical phenomena across different variables, the variables being the aforementioned ones and a physically interpretable model for potential refractivity similar to the one introduced in (Bean and Dutton, 1966; de la Torre Juárez et al.,
2018, e.g.). This analysis also looks at how the vertical integral of $\Delta\Phi$ relates to total column water vapor, and how well this confirms results and observations from prior studies. We explore if vertical profiles of $\Delta\Phi$ and refractivity can help to distinguish possible thermodynamic states and even the contributions from ice vs. liquid water precipitation. Through new statistical and graphical analyses, this study hopes to help understand and quantify these relationships.

      To this end, in Sect. 2, we describe the dataset; in Sect. 3, we outline how we classify different thermodynamic states from
refractivity profiles alone and provide an overview of how we apply k-means clustering to different variables; in Sect. 4, we use our cluster analysis to search for a classification of disparate vertical structures and cross-correlate interpretations of clusters for different variables; and in Sect. 5, we summarize the aims and results of our study.

## 2   Data

The two datasets analyzed and used to train the data classification and potential refractivity model are Level 1C Global Precip-
itation Measurement (GPM) satellite data from the NASA Goddard Space Flight Center and Level 2 Radio Occultations and Heavy Precipitation data from the PAZ satellite (ROHP-PAZ) (Cardellach et al., 2019). From the former, we obtain the Liquid Water Path (LWP, kg/m$^2$) and Ice Water Path (IWP, kg/m$^2$) using Emissivity Principal Component (EPC) profiling retrievals at each pixel across the scan of the GPM passive microwave (PMW) satellite radiometer. The EPC data have a spatial resolution of $0.1° \times 0.1°$, temporal resolution of 30 minutes, and 0.25-km height levels, as described in Appendix A of Turk et al. (2018)
and in Table 1 and Sect. 3 of Turk et al. (2021). Meanwhile, ROHP-PAZ gives refractivity ($N$-units) and $\Delta\Phi$ (mm), all as functions of height at different latitudes, longitudes, and times. Using refractivity and assuming temperature from NCEP, the direct method (Hajj et al., 2002) is applied to derive the pressure (hPa) and water vapor pressure (hPa), whence the temperature (K) used in this study is derived from Eq. (1). For further details on how the aforementioned variables are retrieved from the datasets, we refer the reader to Turk et al. (2021) and the references therein for the GPM dataset and Cardellach et al. (2019)
for the ROHP-PAZ dataset.

      The direct method relies on the ancillary model refractivity agreeing with that of the RO, and with its having the correct temperature distribution for that refractivity profile. While an agreement between the ancillary model and observation should

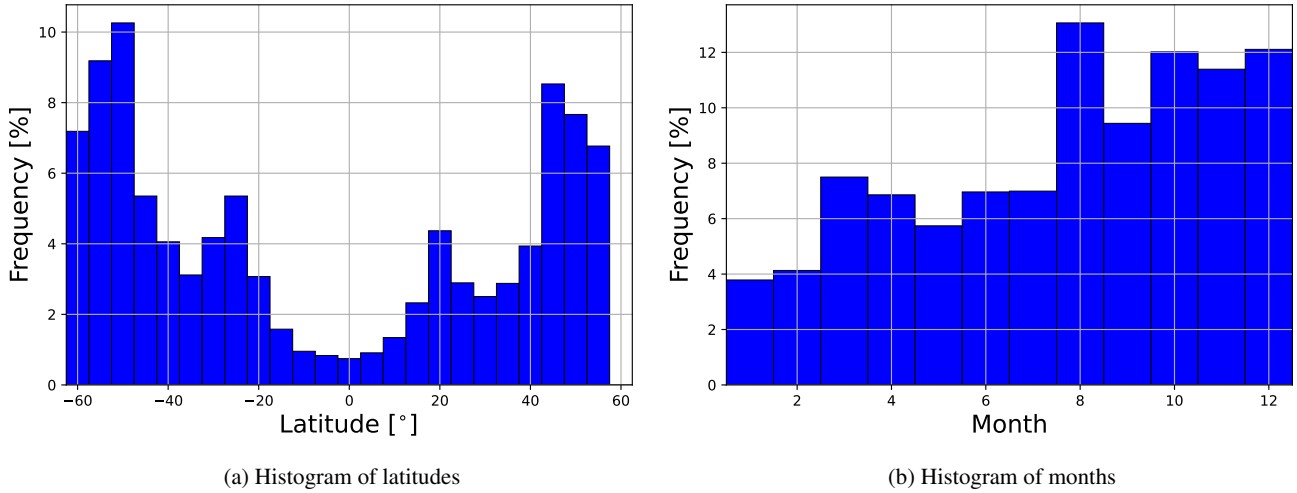

(a) Histogram of latitudes             (b) Histogram of months

**Figure 1.** Sampling distributions for the collocations between the GPM and ROHP-PAZ datasets at different a) latitudes and b) months.

yield reliable retrieved values, using the model temperature profile may introduce retrieval errors when the ancillary model and RO profile refractivities differ significantly (due to, e.g., collocation errors, RO bias, or limited model resolution). The direct

method results in negative or otherwise unrealistic water vapor pressure values serve as quality control flags.

The GPM and ROHP-PAZ profiles are matched across different latitudes, longitudes, and times whenever they coincided within a given spatiotemporal range. As described in Sect. 2 of Turk et al. (2021), the collocation criteria were that the GPM PMW satellite overpass had to occur within $\pm 15$ minutes of the ROHP-PAZ observation, and the ROHP-PAZ observation location had to fall within the PMW satellite's swath. For each ROHP-PAZ observation, the tangent point—the point closest to

the Earth's surface along the ray path—was selected from the lowest level RO. This gives

- 2362 coincidences from July 26th, 2018 to December 31st, 2018 (inclusive);

- 2943 coincidences from March 1st, 2019 to December 31st, 2019;[1] and

- 1401 coincidences from January 1st, 2020 to August 22nd, 2020;

thereby yielding a total of 6706 coincidences from July 26th, 2018 to August 22nd, 2020. At the latitude and longitude of each

coincidence, the collocated data are interpolated onto a grid with equally spaced height intervals of 0.1 km.

Most of the aforementioned coincidences lie poleward of $40°$ N or S, as shown in Figure 1(a), enabling good statistics in those regions. There was a low number of coincidences in the tropics (within $15°$ of the equator) which constrains our analysis in low-latitude regions. Furthermore, as Figure 1(b) shows, there is also a slightly higher number of coincidences in the last four months of the year vs. the first eight, but this poses less of a problem as we do not assess seasonality.

---

[1]Technical issues with the processing of ROHP-PAZ retrievals were encountered in January and February 2019. Although these issues have since been resolved in the currently available ROHP-PAZ dataset, the collocated dataset described in Turk et al. (2021) and analyzed in this study was created before then.

Turk et al. (2021) computed LWP and IWP by integrating the condensed water content (kg/m$^3$)—estimated via EPC passive microwave precipitation profiling (Turk et al., 2018; Utsumi et al., 2020)—along each RO ray path in the ROHP dataset coinciding with GPM data. Integrating the condensed water content along the ray paths ensures that their values are related to $\Delta\Phi$, which is also computed by integrating along each ray path. As a first approximation, we partition the integrated water content into LWP and IWP based on whether the retrieved or model temperature is above or below 273 K, respectively (Turk et al.,

2021). Since non-precipitating supercooled water is not expected to be asymmetric, it should induce little to no polarimetric phase difference (Padullés et al., 2023; Hotta et al., 2024). Hence, this approach misclassifies some supercooled water as ice, in which case this misclassification would predict $\Delta\Phi$ associated with ice when no $\Delta\Phi$ is measured. Finally, as with $\Delta\Phi$, the values of the LWP and IWP at a given latitude, longitude, and height are given according to where the lowest level tangent point for the given ray path lies.

To compute the total column water and ice paths from the aforementioned data for each profile, the water and ice paths are integrated, respectively, from 1 km to 10 km only if a profile has data at 1 and 10 km.[2] For computing the vertical integral of $\Delta\Phi$, since the error associated with this variable in the ROHP-PAZ dataset is roughly $\pm 2$ mm at each height, $\Delta\Phi$ is integrated from 2.5 km to 10 km after rounding $\Delta\Phi$ to the nearest multiple of 2 mm if there exist data from 2 to 2.5 km and at 10 km. The latter condition ensures that the endpoints of the integral are correct and we exclude faulty retrievals which tend to deteriorate

near the bottom of the profiles before the data become corrupted or missing.

Finally, for computing the total column water vapor, the water vapor pressure is integrated from 2.5 km and 10 km, excluding profiles that feature no data at 2.5 km or 10 km, negative water vapor pressure values, or unrealistically high water vapor pressure values—these situations are unphysical and likely result from ancillary model or retrieval errors. The profiles with unrealistically high water vapor pressure values were identified by running an initial $k$-means cluster analysis with $k = 8$—

as explained later in Sects. 3.2 and 3.3—but on every water vapor pressure profile in the dataset. For the dataset, only one cluster contained profiles with unrealistically large water vapor pressure values—at least above 250 hPa at some height in all profiles—while the other clusters contained profiles with water vapor pressure values below 150 hPa. Hence, all profiles in the anomalous cluster were excluded from analyses that relied on water vapor pressure, particularly in the final water vapor pressure cluster analysis.

The number of profiles where these conditions were not met are as follows:

- For total column water vapor: 33 profiles (0.49% of all profiles in the dataset);

- For total column water path: 1 profile (0.01%);

- For total column ice path: 6 profile (0.09%);

- For total column water+ice path: 6 profiles (0.09%); and

---

[2]While requiring path data down to 1 km may seem too stringent, requiring this only ends up excluding six profiles at most for both water and ice path, or under 0.1% of all the profiles in the dataset. Thus, it is not too stringent unless these six profiles happen to be rather extreme cases. The analysis at hand aims at finding general trends and associations rather than atypical cases.

– For the vertical integral of $\Delta\Phi$: 923 profiles (13.76%).

For all cases, the integration is implemented in Python using the composite trapezoidal rule (Atkinson, 1988).

## 3  Methods

The PRO observables are $\Delta\Phi$ and refractivity as functions of height, latitude, and longitude. Hence, this study explores how far one can get with PRO observables while remaining as independent from externally derived weather analyses as possible.
To this end, we develop a model for potential refractivity as a function of height assuming a constant lapse rate (which can be non-adiabatic), hydrostatic balance, and a constant water vapor mixing ratio.

### 3.1  Potential refractivity in a polytropic atmosphere

A first classification criterion organizes profiles based on the differences between observed refractivity profiles and those expected for polytropic atmospheres where air can expand and compress with adiabatic and non-adiabatic heat transfer. If an
air parcel moving vertically through the atmosphere follows a polytropic process—a *polytropic atmosphere*—and the ideal gas law holds, then $p/\rho^m$ and therefore $p^{1-m}T^m$ are constant, where $m$ is the polytropic index of the atmosphere.

We define $K \doteq p^{1-m}(z)T^m(z) = p(z_0)^{1-m}T(z_0)^m$ for some reference height $z_0$. In hydrostatic balance, we have $\frac{\partial p}{\partial z} = -\rho g$, and polytropy also implies that $\frac{\partial p}{\partial z} = \frac{\partial(K\rho^m)}{\partial z} = mK\rho^{m-1}\frac{\partial \rho}{\partial z}$. Balancing these two equations necessitates that $-\rho g = mK\rho^{m-1}\frac{\partial \rho}{\partial z}$, and after multiplying both sides by $\frac{m-1}{mK\rho}$, one gets

$$g\frac{1-m}{mK} = \frac{\partial \rho^{m-2}}{\partial z}.$$

At constant $m \neq 0, 1$, the solution is

$$\rho^{m-1}(z) = \rho(z_0) + g\frac{1-m}{m}(z - z_0). \tag{2}$$

At $m = 0$, the pressure is constant and cannot satisfy hydrostatic equilibrium unless $\rho = 0$, while at $m = 1$, the density decays exponentially, typical of an isothermal atmosphere. When $m = \gamma$, where $\gamma = 5/3$ is the adiabatic index, the change of temper-
ature incurred by air parcels moving vertically in this atmosphere follows an adiabatic process—an *adiabatic atmosphere*.

Using Eq. (2) for the vertical profile of an ideal gas, where $\frac{p(z)}{\rho(z)} = RT(z)$, and by polytropy again, $p(z) = K\rho^m(z) = K\rho(z)\rho^{m-1}(z)$ implies that

$$RT(z) = \frac{p(z)}{\rho(z)} = K\rho^{m-1}(z) = K\left[\rho(z_0) + g\frac{1-m}{m}(z - z_0)\right].$$

This shows that an ideal gas atmosphere in hydrostatic equilibrium and with constant polytropic index $m \neq 0$ with height has
a linear temperature profile $T(z) = \widehat{T}(z_0) - \widehat{\Gamma}(z - z_0)$, where $\widehat{T}(z_0) = \frac{K\rho(z_0)}{R}$ and $\widehat{\Gamma} = -Kg\frac{m-1}{mR}$. When $m = 1$, the solution holds with $\widehat{\Gamma} = 0$ and a constant temperature with height. At constant $m$ and $R$, $\widehat{\Gamma} = -\frac{\partial T}{\partial z}$, and hence, the lapse rate is constant.

When including water vapor processes, one can characterize the temperature profiles in a polytropic atmosphere as 1) a completely dry atmosphere or 2) an *unsaturated* moist atmosphere. Additionally, one can approximate temperature via a

linear relationship with height for 3) a *saturated* moist atmospheric layer where the expansion and contraction of air are reversible, or 4) an atmosphere in which water that condenses in an air parcel is instantaneously removed via precipitation—a pseudoadiabatic atmosphere (e.g., Emanuel, 1994). The lapse rate, $\Gamma$, is nearly constant and called a dry adiabatic lapse rate in the first case, a moist-unsaturated adiabatic lapse rate in the second, a reversible moist-adiabatic lapse rate in the third, and a pseudoadiabatic lapse rate in the fourth. The temperature profile is precisely linear with height for only the first case and close to linear in the others. Each of the four thermodynamic cases above would be represented by a different conservation law (Emanuel, 1994): dry adiabatic (for 1), moist adiabatic (for 2 and 3), pseudoequivalent potential temperatures (for 4), and, by analogy, via a different type of potential refractivity profile. These conserved quantities can be used to define different types of potential refractivity, $\widehat{N}$, based on fitting data to physical laws describing adiabatic and pseudoadiabatic processes (e.g. de la Torre Juárez et al., 2018).

$\widehat{N}$ is derived here for an atmosphere with the following properties: 1) Eq. (1); 2) the ideal gas law; 3) a linear temperature profile with height representative of a polytropic atmosphere; 4) a constant specific humidity representative of a subsaturated atmosphere; and 5) in hydrostatic equilibrium. Deviations between the measured refractivity $N$ and the fit to the model $\widehat{N}$ signal the presence of changes in mixing ratio, precipitation, or non-equilibrium physics (e.g., gravity waves or turbulence). From the above assumptions, we derive in Appendix A the model for $\widehat{N}$:

$$\widehat{N}(z) = \frac{N(z_0)}{[1-c_1(z-z_0)]^2} \times \left\{ (1-c_2)\left[1-c_1(z-z_0)\right]^{c_0} + c_2 \right\}, \tag{3}$$

where $c_0 = \frac{g}{R\widehat{\Gamma}} + 1 = \frac{2m-1}{K(m-1)}$, $c_1 = \frac{\widehat{\Gamma}}{\widehat{T}_0} = \frac{g(m-1)}{m\rho(z_0)}$, and $c_2 = \frac{k_2\widehat{e}_0}{N(z_0)\widehat{T}_0^2}$ are coefficients which must be fit to a given refractivity profile and provide information about the polytropic index. $\widehat{T}_0 \doteq T(z_0)$ & $\widehat{e}_0 \doteq e(z_0)$ are the temperature and water vapor pressure, respectively, at reference height $z_0$. In particular, for $m = 1$, $\widehat{N}$ has an exponential relationship with $z$ (e.g. Bean and Dutton, 1966). The fit coefficients $\mathbf{c} = (c_0, c_1, c_2)$ are defined in terms of the following physical parameters: the acceleration due to gravity on Earth $g = 9.81 \text{ m} \cdot s^{-2}$, specific gas constant of dry air $R = 287.05 \text{ J} \cdot \text{kg}^{-1} \cdot \text{K}^{-1}$, mean tropospheric lapse rate $\widehat{\Gamma}$ (in $\text{K} \cdot \text{km}^{-1}$), and constants $k_1$ and $k_2$ defined in Sect. 1.

The lapse rate can change with height across moist and dry sections, e.g., in the transition between the boundary layer and the free atmosphere (von Engeln et al., 2005; Ao et al., 2012), in the transition from the mid-troposphere to the tropical tropopause layer (TTL) (Fueglistaler et al., 2009), or when clouds are present in a real atmosphere (e.g., Peng et al., 2006; Mascio et al., 2021). Based on these three examples, the fits for $\widehat{N}$ are made in an altitude range that is likely to have a constant lapse rate under the five assumed properties from the last subsection. We set the lower limit of the fit at $z_0 = 2.5$ km, which is mostly above the boundary layer. Furthermore, we set the upper limit to ensure that the fit remains below changes of sign in the lapse rate, such as those caused by gravity waves, stratospheric intrusions, and thermal inversions. Specifically, the upper limit to the fit is 200 m below the lowest height above 5 km where $\left|\frac{\partial T}{\partial z}\right|$ is minimized—which defines a local minimum or a maximum in temperature, as expected for either the cold point tropopause, large gravity waves, or for the bottom of the TTL (Fueglistaler et al., 2009)—and where the temperature is within 10 K of the minimum temperature below 25 km, i.e., within 10 K of the temperature at the cold-point tropopause. Second-order central differences are used to estimate $\frac{\partial T}{\partial z}$ across all of the heights for each given profile (Atkinson, 1988). For this purpose, we use the RO-derived temperature to estimate $T$ to establish a rigorous

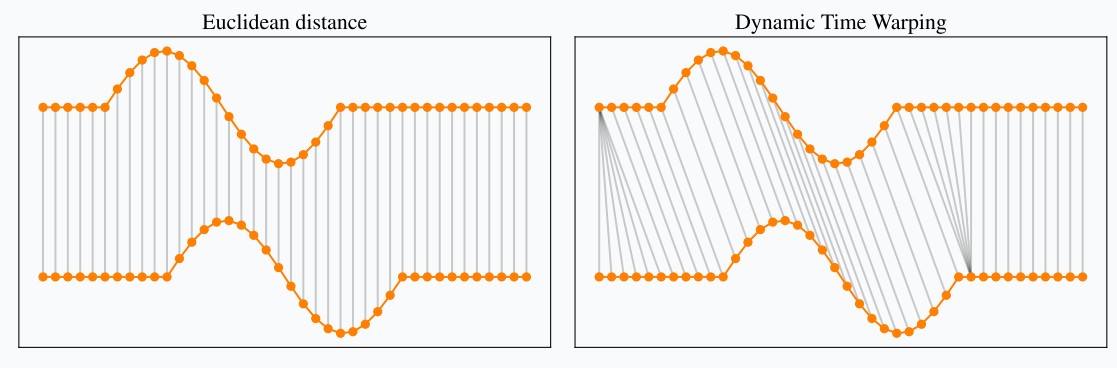

**Figure 2.** A visual comparison (Tavenard, 2021) showing the difference between Euclidean distance and the DTW measure. Time series are shifted vertically in the visualization, but assume that the $y$-axis values match. Thus, between the two time series shown, the Euclidean distance would be nonzero but the DTW measure would be zero.

criterion across all of the profiles and ensure that the fit $\widehat{N}$ is consistently being used where it would be expected to hold, especially for accurate clustering in $N - \widehat{N}$.

More details on the numerical fitting of Eq. (3) are given in Appendix B.

### 3.2 Time series k-means clustering

Across the profiles in the merged dataset described above, we apply $k$-means clustering with $k = 8$ clusters for each of the following variables:

 – RO measured variables: $\Delta\Phi$ (2.5 to 10 km), $N - \widehat{N}$ (2.5 to 8 km), the three fit coefficients for $\widehat{N}$ (the vector **c**), and

240 – Variables from ancillary data: RO+model-derived water vapor pressure ($e$, 2.5 to 10 km) and GPM+RO ray path computed liquid water path (LWP, 1 to 10 km), ice water path (IWP, 1 to 10 km), and total (liquid+ice) water path (TWP, 1 to 10 km).

In all cases aside from the clustering for coefficients (in which case we use standard $k$-means clustering with the standard Euclidean distance), a variation of naive $k$-means clustering called time series $k$-means with dynamic time warping (DTW) (Izakian et al., 2015) is applied. As with naive $k$-means, the dataset is partitioned into $k$ clusters, but instead of measuring the
245 distances between profiles using the Euclidean distance, DTW is used. The numerical procedure for running $k$-means clustering on the aforementioned variables is described in Appendix C.

  We introduce quality-control criteria for each of the variables informed by how $k$-means clustering detected outliers and other physical considerations. For instance, for the $\Delta\Phi$ clustering, we excluded $\Delta\Phi$ profiles where the retrieval for $\Delta\Phi$ cut off
above 2 km, and for the $e$ clustering, we excluded $e$ profiles with unphysically high values, identified using the same clustering technique described in Sect. 2 for excluding $e$ profiles used in the total column water vapor calculations. Including these faulty profiles affects the accuracy when we compare the shapes of the $\Delta\Phi$ profiles for clustering and compute the integral of $\Delta\Phi$.

We found that faulty retrievals tend to deteriorate near the bottom of the profiles before the data become corrupted or missing. The percent of profiles excluded ranged from $0.01\%$ (for LWP clustering) to $13.76\%$ (for $\Delta\Phi$ clustering). See Appendix D for more details on the precise quality-control criteria used for each clustering variable.

### 3.3 Dynamic time warping

DTW is a technique originating in time series analysis that measures the similarity between two signals which are functions of time (or some analogous variable—in this case, height) by finding an optimal alignment between the two signals by "warping" the sample points of each signal such that the measurements in each signal are matched to their nearest point(s) in the other signal as measured by the Euclidean norm, regardless of the times at which each point was measured (Müller, 2007). We still assume that the start and end points match in each case, that the ordering of measurements (with time) within each profile stay the same, and that each point in one signal is matched to at least one point in the other. This ensures the following:

1. For cases of missing or uneven data points within a given profile, we can still compare the rough shape of this profile with others, and

2. For translations in sampling (e.g., when two measurements are out of phase or when recorded heights are imprecise), DTW can make up for this by shifting the heights at which measurements are taken when comparing two profiles.

See Tavenard (2021) or Müller (2007) for more details on how DTW is calculated.

Figure 2 features an intuitive visualization of how DTW works when comparing time series. The featured example is taken from Tavenard (2021) and shows two signals consisting of horizontal lines combined with one period of a sinusoid. Note how DTW matches the patterns and overall shape of each time series, which intuitively should result in a more sound similarity assessment than when using the Euclidean distance, since the latter matches timestamps (or heights for this study) regardless of when they were sampled.

## 4   Results and analysis

Clustering provides an initial classification for the types of atmospheric profiles that can occur across the dataset by looking at the centroids in different clustering variables. Figure 3 shows the results of $k$-means cluster analysis when applied to the following variables in the dataset: (a) $N - \widehat{N}$, (b) $\Delta\Phi$, (c) IWP, (d) LWP, (e) TWP, and (f) water vapor pressure. The plots in Fig. 3 show the eight clustering centroids for each variable. Each centroid is an average profile representing the general shape and magnitude of the indicated variable for profiles within its cluster.

A second step in the analysis uses frequency histograms of different cluster groups to summarize the relationships between clusters. Tables 1, 2, and 3 are frequency histograms that compare clustering in different variables: $N - \widehat{N}$ with water vapor pressure, the $\widehat{N}$ coefficients **c** with water vapor pressure, and $\Delta\Phi$ against the path variables (LWP, IWP, and TWP), respectively. These tables look for patterns in the ability of $N - \widehat{N}$ to predict different distributions of vertical water vapor pressure and $\Delta\Phi$ to predict different types of water path profiles across the vertical profiles in the dataset. Percentages in the topmost row and

**Table 1.** Percentage of profiles in each $e$ cluster (column) for each $N - \widehat{N}$ cluster (row). Cluster numbers are ordered from smallest (most negative/zero) to largest (most positive) value by comparing their corresponding centroids in Fig. 3. Overbars and underbars indicate percentages that fall above or below, respectively, 1.5 times the weighted standard deviation (STD) from the mean percentage for each given row; the STD is weighted by the percentage of $N - \widehat{N}$ corresponding to each case (6.66%, 5.34%, 7.06%, etc.). Bolded black and bolded grey indicate the maximum and minimum percentages for each row, respectively.

$N - \widehat{N}$: most negative … most positive →

$e$: driest … wettest ↓

| $N - \widehat{N}$: | → | 6.66% | 5.34% | 7.06% | 16.97% | 3.75% | 38.69% | 18.12% | 3.41% |
|---|---|---|---|---|---|---|---|---|---|
| $e$: | ↓ | 3 | 2 | 6 | 8 | 7 | 4 | 1 | 5 |
| 21.35% | 3 | **0.00%** | 0.29% | **0.00%** | 5.09% | **0.00%** | 30.86% | **41.19%** | 0.90% |
| 20.53% | 7 | **0.46%** | 1.73% | 0.66% | 14.45% | 2.88% | 27.63% | **31.66%** | 23.98% |
| 19.63% | 2 | 8.56% | 10.12% | **0.66%** | **35.45%** | 8.23% | 23.05% | 14.81% | 23.53% |
| 12.87% | 8 | 14.35% | 24.57% | **2.18%** | 21.91% | 12.76% | 11.12% | 7.15% | **28.51%** |
| 10.89% | 6 | 30.09% | **32.66%** | 7.42% | 15.91% | 29.22% | 4.90% | **3.91%** | 14.93% |
| 7.27% | 4 | **30.56%** | 20.52% | 19.43% | 6.09% | 24.28% | 1.59% | **0.60%** | 7.69% |
| 6.16% | 1 | 14.35% | 8.96% | **55.46%** | 0.82% | 18.52% | 0.36% | 0.09% | **0.00%** |
| 1.29% | 5 | 1.16% | 0.87% | **14.19%** | 0.09% | 3.70% | **0.00%** | **0.00%** | 0.45% |

**Table 2.** Percent of profiles in each $e$ cluster (column) for each $\mathbf{c}$ cluster (row). $e$ cluster numbers are ordered roughly from smallest to largest value by comparing their corresponding $e$ centroids in Fig. 3(f) while the $\mathbf{c}$ cluster numbers are merely listed in numerically increasing order (arbitrarily). Bolding, coloring, and barring are for each row as in Table 1.

$e$: driest … wettest ↓

| $\mathbf{c}$: | → | 21.73% | 24.13% | 16.19% | 19.89% | 7.60% | 0.07% | 4.82% | 5.57% |
|---|---|---|---|---|---|---|---|---|---|
| $e$: | ↓ | 1 | 2 | 3 | 4 | 5 | 6 | 7 | 8 |
| 21.35% | 3 | 3.37% | **43.54%** | 34.65% | 11.55% | **2.75%** | 40.00% | 34.67% | 3.75% |
| 20.53% | 7 | 9.34% | 29.13% | 29.49% | 18.98% | 8.06% | **0.00%** | **35.91%** | 8.85% |
| 19.63% | 2 | 19.78% | 16.57% | 15.21% | **27.23%** | 22.59% | **0.00%** | 15.79% | 16.09% |
| 12.87% | 8 | 15.73% | 6.12% | 9.31% | 16.05% | 22.40% | **40.00%** | **6.50%** | 21.18% |
| 10.89% | 6 | 17.17% | 2.66% | 6.54% | 11.18% | **27.11%** | 20.00% | **1.86%** | 18.50% |
| 7.27% | 4 | **16.21%** | 1.24% | 3.13% | 6.30% | 12.57% | **0.00%** | 1.55% | 11.26% |
| 6.16% | 1 | 15.11% | 0.37% | 0.46% | 6.98% | 3.34% | **0.00%** | 0.62% | **18.23%** |
| 1.29% | 5 | **3.09%** | 0.19% | 0.65% | 1.58% | 0.20% | **0.00%** | 0.62% | 1.88% |

leftmost column reflect the total number of profiles that meet the clustering requirements for each specified clustering variable. Green and red indicate the maximum and minimum percentages within each respective row. Note that since profiles were excluded from the cluster analyses for certain variables, the weighted averages for each column or row will not always add up as expected from the law of total probability.

### 4.1 Total column $\Delta\Phi$ and total column water vapor

Bretherton et al. (2004) showed an exponentially increasing relationship between precipitation and total column relative humidity over the tropics. Later studies (Muller et al., 2009; Holloway and Neelin, 2010; Emmenegger et al., 2022) demonstrate

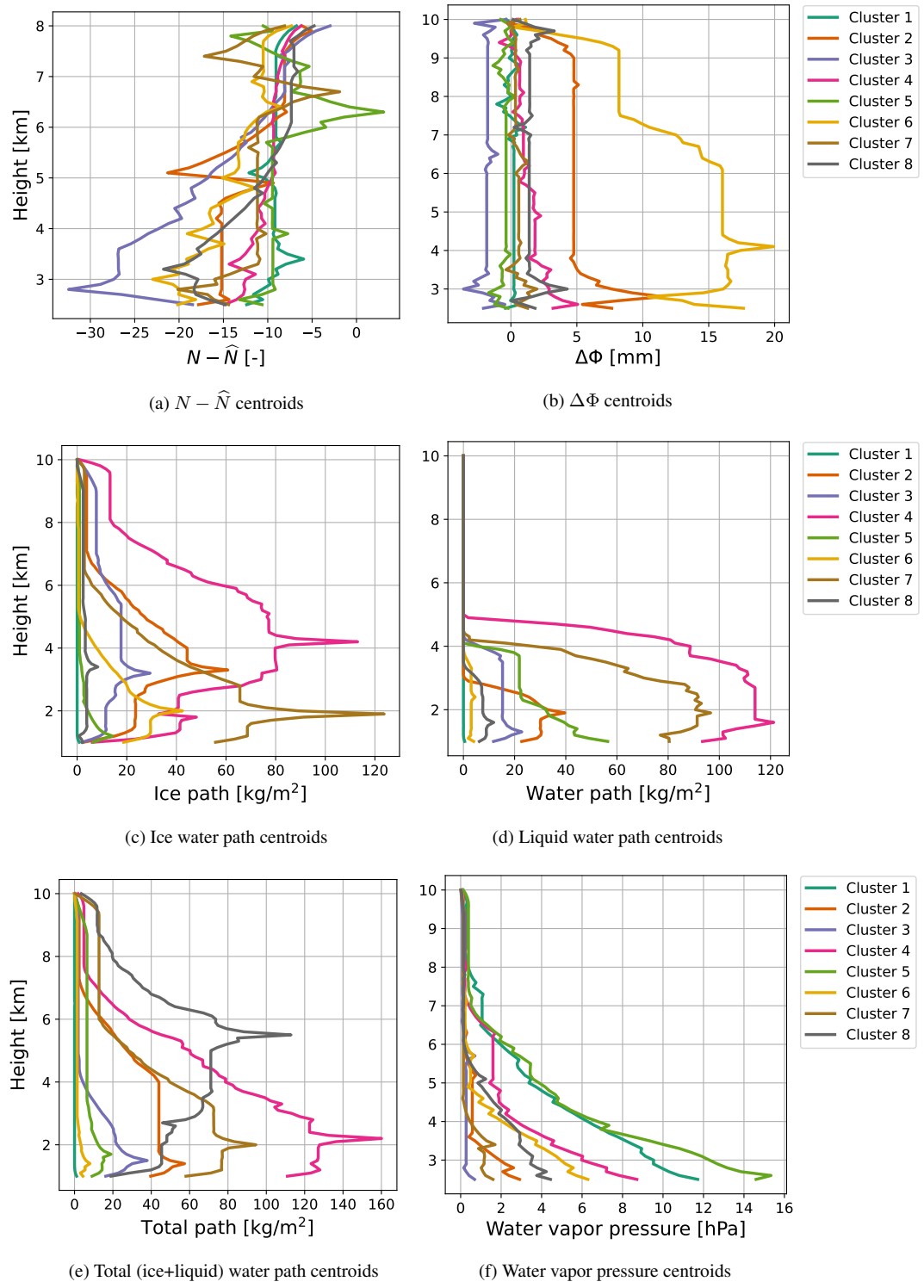

(a) $N - \widehat{N}$ centroids

(b) $\Delta\Phi$ centroids

(c) Ice water path centroids

(d) Liquid water path centroids

(e) Total (ice+liquid) water path centroids

(f) Water vapor pressure centroids

**Figure 3.** Cluster analysis centroids computed by applying time series $k$-means clustering across all variables of interest (excluding the $\widehat{N}$ coefficients **c**).

**Table 3.** Percent of profiles in each cluster for the column variable listed—liquid water path (LWP), ice water path (IWP), and liquid+ice water path (TWP), respectively— for each ΔΦ cluster indicated by the row. Cluster numbers are ordered from smallest (most negative/zero) to largest (most positive) value by comparing their corresponding centroids in Fig. 3. Bolding, coloring, and barring are for each row as in Table 1.

ΔΦ: most negative/zero  …  most positive

LWP: driest … wettest

| ΔΦ: → LWP: ↓ | | 5.20% 3 | 25.32% 5 | 34.62% 1 | 23.45% 7 | 3.87% 8 | 5.74% 4 | 1.56% 2 | 0.24% 6 |
|---|---|---|---|---|---|---|---|---|---|
| 89.14% | 1 | **96.35%** | 95.70% | 95.25% | 91.22% | 66.07% | 54.52% | 17.78% | 7.14% |
| 5.34% | 6 | 3.32% | 3.28% | 3.15% | 6.05% | 11.61% | **15.96%** | 6.67% | 0.00% |
| 2.85% | 8 | 0.33% | 0.75% | 1.25% | 1.99% | 12.05% | 15.96% | 7.78% | **28.57%** |
| 1.51% | 3 | 0.00% | 0.20% | 0.15% | 0.44% | 8.04% | 11.14% | **22.22%** | 7.14% |
| 0.81% | 2 | 0.00% | 0.00% | 0.20% | 0.22% | 2.23% | 2.11% | 27.78% | **28.57%** |
| 0.28% | 5 | 0.00% | 0.00% | 0.00% | 0.07% | 0.00% | 0.30% | **14.44%** | 14.29% |
| 0.04% | 7 | 0.00% | 0.00% | 0.00% | 0.00% | 0.00% | 0.00% | 2.22% | **7.14%** |
| 0.03% | 4 | 0.00% | 0.00% | 0.00% | 0.00% | 0.00% | 0.00% | 1.11% | **7.14%** |

ΔΦ: most negative/zero  …  most positive

IWP: driest … wettest

| ΔΦ: → IWP: ↓ | | 5.20% 3 | 25.32% 5 | 34.62% 1 | 23.45% 7 | 3.87% 8 | 5.74% 4 | 1.56% 2 | 0.24% 6 |
|---|---|---|---|---|---|---|---|---|---|
| 85.78% | 1 | **94.68%** | 94.26% | 93.91% | 85.03% | 63.39% | 29.82% | 3.33% | 0.00% |
| 7.31% | 8 | 2.99% | 4.78% | 4.80% | 9.88% | 18.75% | **19.58%** | 3.33% | 7.14% |
| 3.75% | 5 | 1.99% | 0.75% | 0.85% | 4.06% | 8.93% | **32.23%** | 10.00% | 14.29% |
| 1.72% | 3 | 0.33% | 0.14% | 0.35% | 0.88% | 7.14% | 13.25% | **21.11%** | 7.14% |
| 0.75% | 6 | 0.00% | 0.00% | 0.05% | 0.07% | 0.00% | 4.22% | **31.11%** | 14.29% |
| 0.49% | 2 | 0.00% | 0.00% | 0.05% | 0.00% | 1.34% | 0.90% | 22.22% | **28.57%** |
| 0.13% | 4 | 0.00% | 0.00% | 0.00% | 0.07% | 0.45% | 0.00% | **7.78%** | 0.00% |
| 0.07% | 7 | 0.00% | 0.00% | 0.00% | 0.00% | 0.00% | 0.00% | 1.11% | **28.57%** |

ΔΦ: most negative/zero  …  most positive

TWP: driest … wettest

| ΔΦ: → TWP: ↓ | | 5.20% 3 | 25.32% 5 | 34.62% 1 | 23.45% 7 | 3.87% 8 | 5.74% 4 | 1.56% 2 | 0.24% 6 |
|---|---|---|---|---|---|---|---|---|---|
| 84.55% | 1 | 93.02% | 92.96% | **93.26%** | 84.14% | 60.71% | 28.92% | 3.33% | 0.00% |
| 8.46% | 6 | 4.98% | 5.81% | 5.09% | 12.09% | 15.63% | **24.40%** | 3.33% | 0.00% |
| 3.99% | 5 | 1.99% | 1.09% | 1.15% | 2.88% | 17.41% | **29.82%** | 12.22% | 14.29% |
| 1.79% | 3 | 0.00% | 0.07% | 0.45% | 0.66% | 5.36% | 14.46% | **25.56%** | 7.14% |
| 0.82% | 2 | 0.00% | 0.00% | 0.05% | 0.15% | 0.45% | 2.41% | 35.56% | **42.86%** |
| 0.22% | 7 | 0.00% | 0.00% | 0.00% | 0.00% | 0.45% | 0.00% | 12.22% | **14.29%** |
| 0.09% | 8 | 0.00% | 0.00% | 0.00% | 0.07% | 0.00% | 0.00% | 4.44% | **7.14%** |
| 0.07% | 4 | 0.00% | 0.00% | 0.00% | 0.00% | 0.00% | 0.00% | 3.33% | **14.29%** |

a similar and related positive relationship between precipitation and total column water vapor (TCWV) in the tropics, where under a certain TCWV value, precipitation is generally near-zero in a given profile, and above a "pickup" threshold in TCWV,

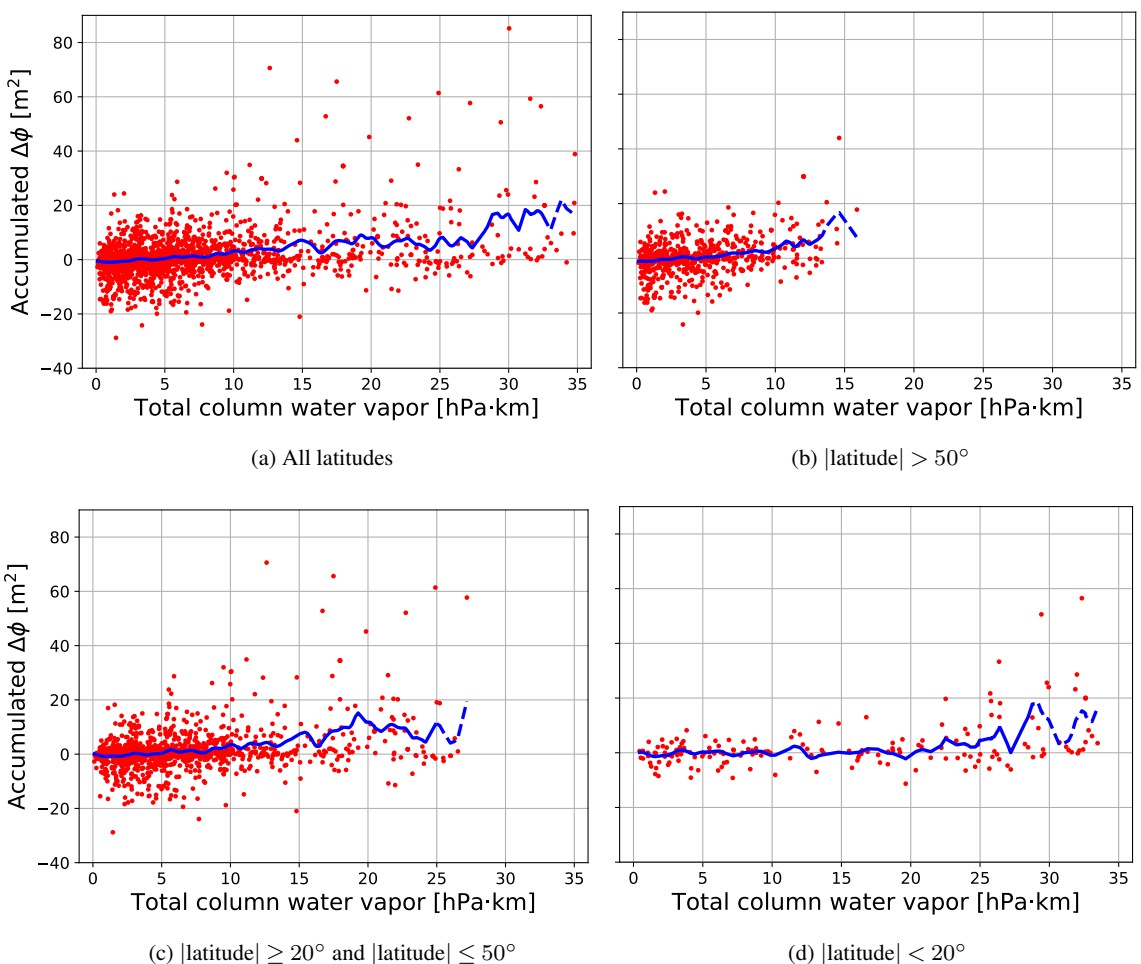

(a) All latitudes

(b) $|\text{latitude}| > 50°$

(c) $|\text{latitude}| \geq 20°$ and $|\text{latitude}| \leq 50°$

(d) $|\text{latitude}| < 20°$

**Figure 4.** Moving averages (blue) of accumulated $\Delta\Phi$ vs. accumulated water vapor pressure over scatter plots (red) across all latitudes and in different latitudinal ranges. Dashed portions of the moving averages correspond to where each bin had less than $34$ data points.

**Table 4.** Pearson's correlation coefficient ($r$), Spearman's rank correlation coefficient ($\rho$), and Kendall's rank correlation coefficient ($\tau$) on all pairs of the accumulated $\Delta\Phi$ vs. total column water vapor for the raw dataset (Table 4(a)) and for the moving averages (Table 4(b)) across varying latitudinal ranges. Each correlation coefficient has a $p$-value below $10^{-9}$, indicating a high statistical significance for all coefficients.

| Lat. range: $\rightarrow$ | All | $> 50°$ | $\geq 20°$ and $\leq 50°$ | $< 20°$ |
|---|---|---|---|---|
| Cor. coef.: $\downarrow$ | | | | |
| Pearson's $r_p$ | 0.332 | 0.315 | 0.349 | 0.375 |
| Spearman's $\rho_s$ | 0.216 | 0.223 | 0.206 | 0.287 |
| Kendall's $\tau_k$ | 0.147 | 0.151 | 0.139 | 0.194 |

(a) Correlation tests on the raw dataset

| Lat. range: $\rightarrow$ | All | $> 50°$ | $\geq 20°$ and $\leq 50°$ | $< 20°$ |
|---|---|---|---|---|
| Cor. coef.: $\downarrow$ | | | | |
| Pearson's $r_p$ | 0.940 | 0.901 | 0.921 | 0.708 |
| Spearman's $\rho_s$ | 0.971 | 0.964 | 0.947 | 0.683 |
| Kendall's $\tau_k$ | 0.864 | 0.847 | 0.803 | 0.508 |

(b) Correlation tests on the moving averages

precipitation *may* become non-negligible and increase exponentially. To evaluate the statistical representativity of our dataset, we tested the validity of using the magnitude of $\Delta\Phi$ as a proxy for the magnitude of precipitation by looking for a monotonic relationship—and in particular, the precipitation pickup pattern (Holloway and Neelin, 2010)—between TCWV and the total column of the PRO observable $\Delta\Phi$.

Figure 4 presents scatter plots of accumulated $\Delta\Phi$ vs. TCWV for all profiles in the dataset at (a) all latitudes, (b) upper mid-latitudes (above $50°$), (c) subtropics and mid-latitudes (between $20°$ and $50°$), and (d) tropics (below $20°$), with overlaid moving averages. These moving averages were done using the `filter1d` tool Generic Mapping Tools (`gmt`) Version 6.3 (Wessel et al., 2019). Averaging was done with a Gaussian filter of width $2\,\mathrm{hPa \cdot km}$ (option `-Fg2`) excluding outputs where the input data has a gap exceeding $0.2$ (option `-L0.2`) and including ends of the time series in the output (option `-E`).

The sparse statistics and high variability across higher moisture profiles within the dataset make it difficult to filter out outlier profiles that could significantly bias the moving averages. Thus, Fig. 4 shows insufficient data for higher moisture cases to replicate the precipitation pickup pattern with much fidelity, as shown by the dashed lines in Fig. 4.[3] Nonetheless, after averaging, a positive correlation between accumulated $\Delta\Phi$ and TCWV was found across *all* latitudes ($r_p = 0.940$) and for the three latitudinal ranges separately; see Table 4(b).

The strength of the correlation between accumulated $\Delta\Phi$ and TCWV also depends on which data the correlation analyses ran. The correlation coefficients in Table 4(a) indicate a low positive correlation between accumulated $\Delta\Phi$ and TCWV in the raw dataset—i.e., on the individual profiles. After applying the Gaussian filter with results in Fig. 4 and running correlation analyses on the filtered data, we find a high positive correlation between the same two quantities in Table 4(b). This suggests that, on average, there is a global positive relationship between the total column $\Delta\Phi$ and water vapor pressure, but this relationship is weak across individual profiles. Hence, when classifying individual profiles, accumulated $\Delta\Phi$ does not appear to be a good proxy for precipitation on a single profile; Sect. 4.2 gives a more useful way to predict water vapor pressure profiles using RO observables.

Similarly, Figures 4(b) and 4(d) show that, even when using running means, the limited data and high variability across individual profiles only weakly suggest a threshold at which TCWV begins to induce precipitation—i.e., the critical level at which the precipitation pickup starts. This threshold appears to be notably lower in the upper mid-latitudes than in the tropics: the accumulated $\Delta\Phi$ moving averages reach similar magnitudes at approximately $12$-$13\,\mathrm{hPa \cdot km}$ in high latitudes vs. $25$-$26\,\mathrm{hPa \cdot km}$ in the tropics. However, particularly for tropical profiles, significantly more data are needed to robustly confirm how accurately polarimetry can capture the precipitation pickup pattern from accumulated $\Delta\Phi$ averages.

## 4.2 $N - \widehat{N}$ and water vapor pressure

We represent the deviations of $N$ from a profile with the properties outlined in Sect. 3.1 by looking at overlaid graphs of $N$ and $\widehat{N}$ as functions of height and by plotting $N - \widehat{N}$ as a function of height. Figure 5 shows two examples—Figs. 5(a) and 5(b)—where $N - \widehat{N}$ does not correlate strongly with $\Delta\Phi$, whereas Fig. 5(c) highlights a profile in which a small bump in $N - \widehat{N}$ and in water vapor correlate with a large $\Delta\Phi$. This supports the interpretation of $\Delta\Phi$ as caused by an ice cloud. Figures 5(a) and

---

[3]$34$ counts per bin was chosen as a consistent threshold for all plots in Fig. 4 to show where the density of data falls below a given reference value.

5(b), instead, demonstrate the ability of the deviation from potential refractivity $N - \widehat{N}$ to predict moisture distributions, even when $\Delta\Phi$ shows little to no correlation with these moisture changes as a function of height. For example, the profile in Fig. 5(b) shows negligible $\Delta\Phi$, suggesting that the water vapor profile likely indicates ice crystal-free clouds from approximately 7.5 km down to near 5.5 km (see, e.g., the method in Peng et al., 2006).

Hence, Fig. 5(b) shows that differences in $N$ from $\widehat{N}$ tend to correspond with altitudinal excursions from a near-exponential water vapor pressure as expected for a constant $c_2$ in Eq. (B). Table 1 verifies this by measuring the frequency with which different $N - \widehat{N}$ clusters agree with specific $e$ clusters; their centroids are shown in Figs. 3(a) and 3(f), respectively. For example, Cluster 1 for $N - \widehat{N}$ is the most flat and occurs most frequently, correlates most strongly, to Clusters 3 and 7 for $e$, the latter of which correspond to profiles with little to no moisture. Conversely, Cluster 6 for $N - \widehat{N}$ correlates well with the highest-moisture profiles in Clusters 1 and 5 for $e$ and contains almost none of the low or no moisture profiles (Clusters 3, 7, 2, and 8 for $e$).

The $N - \widehat{N}$ centroids in Fig. 3(a) tend to deviate from a constant value, primarily in the negative direction for $N - \widehat{N}$ clusters associated with higher moisture, e.g., $N - \widehat{N}$ Cluster 6. This indicates that $N < \widehat{N}$ within a profile correlates with the presence of moisture, as a higher specific humidity generally increases refractivity (Friehe et al., 1975; Takamura et al., 1984, also see Eq. (1)). Hence, because the potential refractivity $\widehat{N}$ is fit across both moist and dry regions of a profile, the background measured refractivity $N$ in regions without moisture may fall below the vertically representative $\widehat{N}$.

On the other hand, as shown in Table 1, Cluster 3 for $N - \widehat{N}$ features larger values of $\left| N - \widehat{N} \right|$ than Cluster 6 for $N - \widehat{N}$ yet does not correlate with profiles that have a higher water vapor pressure (i.e., Clusters 1 and 5 for $e$). The examples in Fig. 5 also demonstrate this; in particular, Fig. 5(c) features a profile with a notably higher value of $e$ than the one in Fig. 5(b) yet exhibits smaller values of $\left| N - \widehat{N} \right|$ overall. This suggests that the actual *magnitude* of deviations of $N$ from $\widehat{N}$ does not necessarily correspond with the magnitude of water vapor pressure. Nonetheless, the clustering indicates a weak inverse relationship between the $N - \widehat{N}$ and $e$—the upper-left and bottom-right corners of Table 1 consist mostly of red values while the bottom-left and upper-right corners consist mostly of green ones.

The aforementioned observation raises two possible hypotheses for why the relationship between the magnitudes of $N - \widehat{N}$ and $e$ are not more direct. Firstly, it is possible that the relationship between $e$ and $N - \widehat{N}$ is between the derivatives of one or both. Furthermore, $\widehat{N}$ is fit to most of the troposphere down to 2.5 km. Hence, the difference between the measured refractivity, $N$, and the potential refractivity model, $\widehat{N}$, is most pronounced when there are concentrated moisture anomalies within narrow bands of the troposphere. Conversely, the sensitivity of the derivatives of $N$ and $\widehat{N}$ with respect to height suggests that there could be cases where a profile is moist, yet the model $\widehat{N}$ still closely matches the observed $N$. This can happen when a moist-unsaturated adiabatic lapse rate (Emanuel, 1994) holds throughout most of the profile. In such cases, $N - \widehat{N}$ could be close to zero, even when the water vapor pressure remains elevated, provided that the water vapor pressure gradients remain small. As an example, the centroid for $N - \widehat{N}$ Cluster 7 is relatively flat (Fig. 3(a)), but Table 1 shows that $e$ Clusters 4 and 6—both moderately high moisture cases (Fig. 3(f))—are the most commonly represented $e$ clusters in $N - \widehat{N}$ Cluster 7.

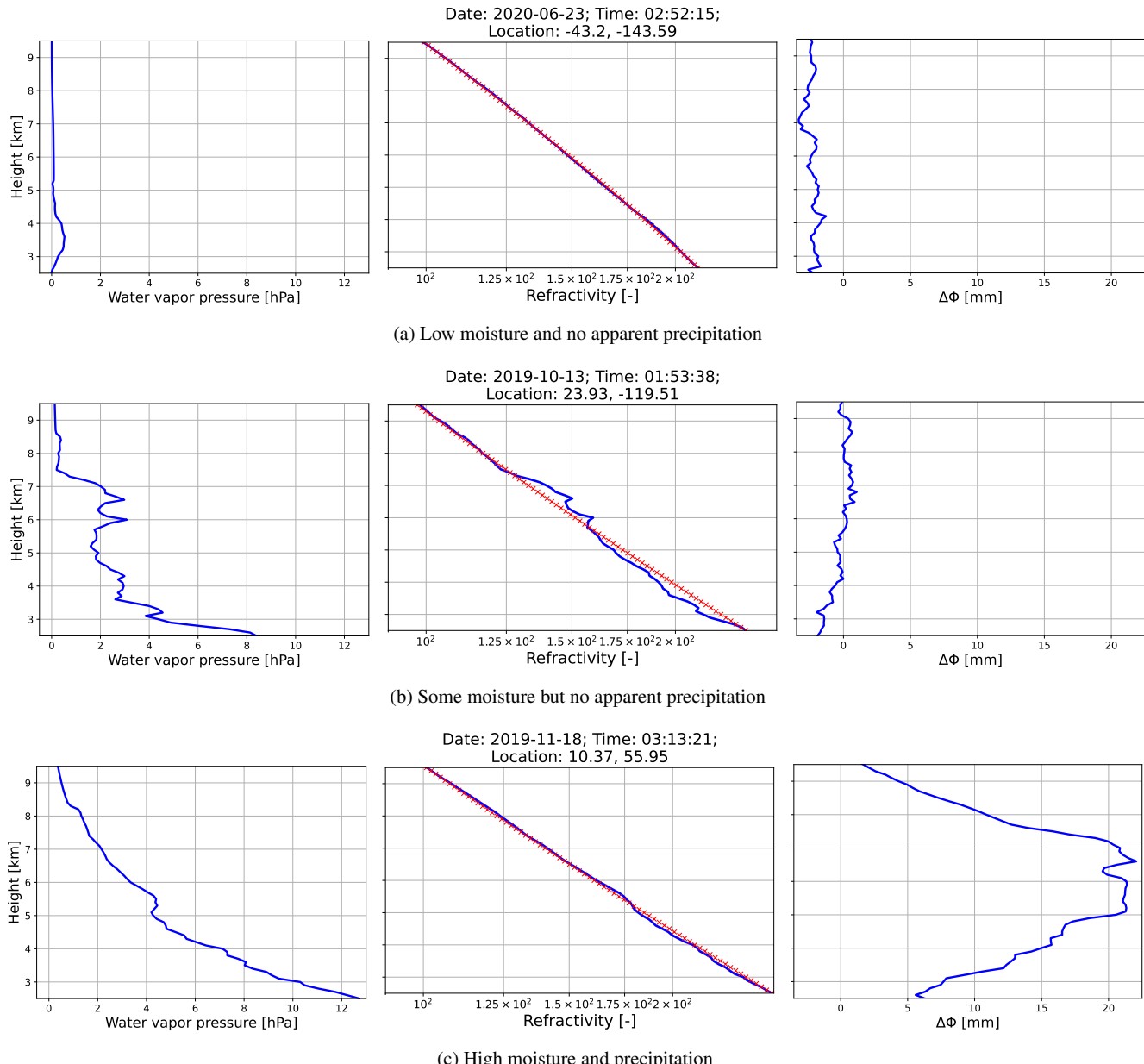

**Figure 5.** Three examples of thermodynamic profiles with different moisture and precipitation contents at various times and locations. For each, we show the height on the $y$-axes and the following on the $x$-axes: $e$ (left); $N$ in blue and $\widehat{N}$ in red (center); and $\Delta\Phi$ (right).

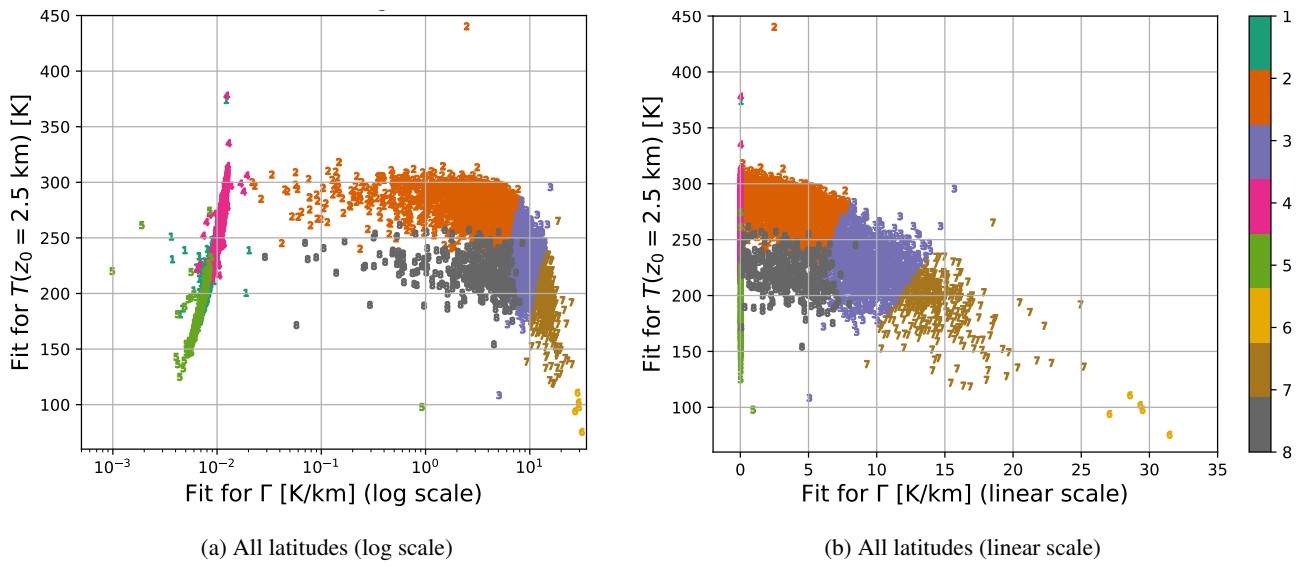

(a) All latitudes (log scale)  (b) All latitudes (linear scale)

**Figure 6.** Scatter plots of the best-fit values of $\widehat{T}_0$ vs. $\widehat{\Gamma}$ across all latitudes in the dataset using a) logarithmic scaling and b) linear in $\widehat{\Gamma}$ to make the separation in $\widehat{\Gamma}$ more apparent for small values of $\widehat{\Gamma}$. The colors and symbols correspond to the associated $\widehat{N}$ coefficient (**c**) clusters for each point, as indicated by the color bars on the right-hand sides.

### 4.3  $\widehat{N}$ model coefficients and cluster groups

The $\widehat{N}$ coefficients $\mathbf{c} = (c_0, c_1, c_2)$ tend to only exhibit two degrees of freedom across the profiles in the dataset. Figure 6 shows how projecting the **c** clusters onto the $\left(\widehat{\Gamma}, \widehat{T}_0\right)$-plane leads to a clear partitioning across different **c** clusters. This suggests that the dominant clusters for $\widehat{e}$ (and therefore $e$) in the dataset are related to changes in $\widehat{\Gamma}$ and $\widehat{T}_0$. Note that changes in $\widehat{\Gamma}$ and $\widehat{T}_0$ are related to changes in the polytropic index $m$ and therewith the underlying heat transfer thermodynamics.

The clustering across **c** was generally able to partition the physical and nonphysical fits. Clusters 2, 3, 7, and 8 for **c** feature

physical values of $\widehat{T}_0$ and $\widehat{\Gamma}$ while the other clusters feature nonphysically extreme values of $\widehat{T}_0$ (mainly Cluster 6), $\widehat{\Gamma}$ (Clusters 1 and 4) or both (Cluster 5). Such nonphysical fits indicate where the assumed physics is not reflective of the actual physics in those profiles. Sometimes, we observed that faulty retrievals fell within these clusters with unphysical profiles, suggesting (and perhaps identifying) retrieval issues—e.g., those discussed in Sect. 2—rather than physical phenomena.

Figure 6 shows a moderately negative linear correlation between $\widehat{T}_0$ and $\widehat{\Gamma}$ for the fits which feature physically realistic values

of $\widehat{\Gamma}$. Between $\widehat{T}_0$ and $\widehat{\Gamma}$ across all latitudes for $\widehat{\Gamma} > 0.1$, we have a Pearson correlation coefficient of $-0.697$, a Spearman rank correlation coefficient of $-0.676$, and a Kendall rank correlation coefficient of $-0.497$. Each correlation coefficient has a $p$-value below machine epsilon (i.e., at least below $2.22 \times 10^{-16}$), thereby showing the statistical significance of this negative correlation. This correlation reflects that the moist adiabatic lapse rate has a negative relationship with temperature for profiles with sufficient moisture. Since the moist adiabatic lapse rate approaches the dry adiabatic lapse rate for temperatures roughly

below 230 K, a higher lapse rate can be observed for colder profiles.

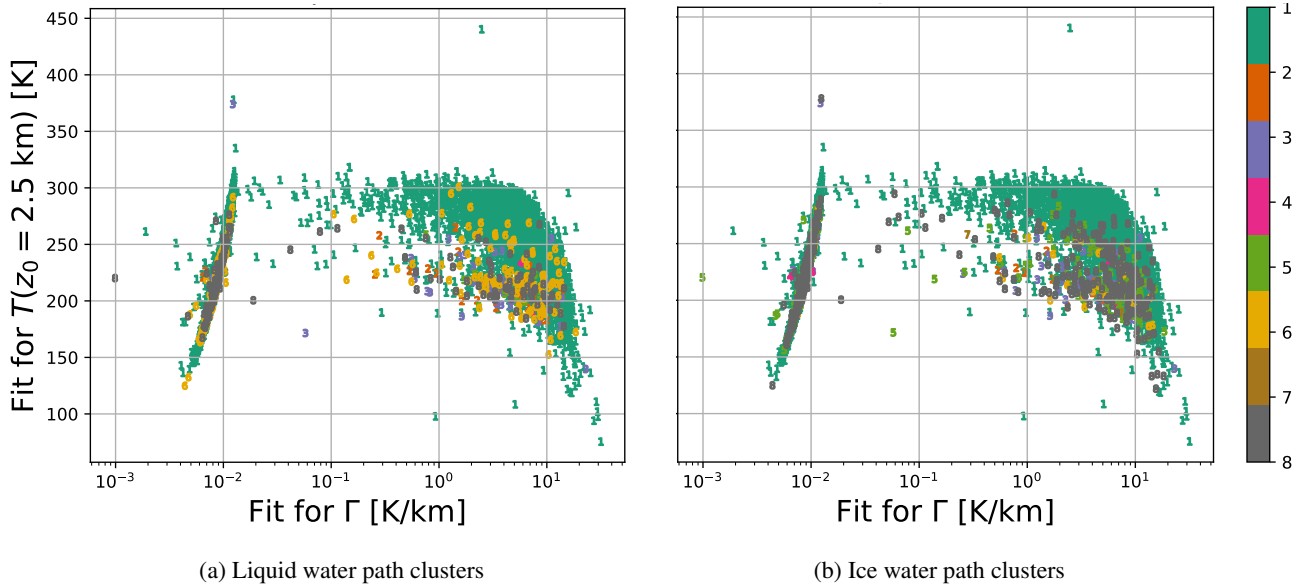

(a) Liquid water path clusters        (b) Ice water path clusters

**Figure 7.** LWP and IWP clusters over $\widehat{T}_0$ vs. $\widehat{\Gamma}$ across all profiles and latitudes.

Figure 7 features two scatter plots that show how the fit coefficient vector **c** relates to the LWP and IWP as a function of height. The fit values of $\widehat{\Gamma}$ and $\widehat{T}_0$ generally do not correlate with path clusters. However, when $\widehat{\Gamma} > 10^{-1}$ K/km and $\widehat{T}_0 > 280K$ for a given profile, that profile has little to no precipitation, as shown by the near-uniformity of Cluster 1 (turquoise) for either LWP or IWP in that region, as indicated by Figs. 7(a) and 7(b), respectively. That is, **c** is not too informative in confirming the presence of precipitation; however, **c** can sometimes rule out the presence of moisture, and thereby precipitation. Similar yet weaker relationships between **c** and particular precipitation regimes can also be seen across other ranges of $\widehat{\Gamma}$ and $\widehat{T}_0$ in Fig. 7, e.g., $\widehat{\Gamma} > 10$ K/km tends to also correlate with low or no moisture cases.

As the aforementioned relationship between $N - \widehat{N}$ and $e$ suggests, **c** also exhibits an apparent relationship with $e$. Figure 6 suggests that Cluster 2 for **c** tends to contain profiles where $\widehat{e}$ is near-zero. This tends to correspond to cases when $e$ is too low for there to be precipitation; as confirmed in Fig. 4, when the water vapor pressure is too low, precipitation cannot form. The relationship between **c** and $e$ may be analyzed more precisely by looking at Table 2, which demonstrates the predictive power in using **c** clusters to predict representative water vapor pressure profiles, i.e., the centroids for $e$ clusters shown in Fig. 3(f).

### 4.4 $\Delta\Phi$ and both liquid & ice water path

Table 3 explores the correlation of PRO $\Delta\Phi$ profiles with precipitation in a given profile. Clusters with large $\Delta\Phi$ tend to correlate with those of large LWP or IWP, and inversely, those with small $\Delta\Phi$ also relate to profiles with little to no LWP and IWP. This should already be expected, as prior studies (e.g., Cardellach et al., 2019; Wang et al., 2022; Padullés et al., 2023) already indicate relationships between $\Delta\Phi$ and both water precipitation and ice.

Despite how Clusters 2 and 6 for $\Delta\Phi$ feature large values of $\Delta\Phi$ ($> 4$ mm) quite deep into the atmosphere—up to around 9 km according to their respective centroids—Table 3 shows that ice precipitation is not necessarily deep for those cases. In particular, Clusters 2 and 6 for $\Delta\Phi$ both correlate well with Clusters 2 and 6 for IWP, but the centroids for the latter two drop to zero near 7 and 5 km, respectively. This could be because $\Delta\Phi$ across different heights need not correspond one-to-one with the LWP nor the IWP at those heights, and also because LWP and IWP do not necessarily signal precipitation right at the time they are measured.

Even though the height of a particular onset or peak in $\Delta\Phi$ might not correlate with onsets or peaks, respectively, in the path cluster centroids, the shapes of the $\Delta\Phi$ and total path cluster centroids appear to correlate in both precipitating and non-precipitation cases, as demonstrated in Table 3. This consistency in shape but not in height is a property of the DTW measure used for the clustering. Hence, the lack of height correlations in our clusters does not contradict the model predictions of Padullés et al. (2023) since their model directly matches features in $\Delta\Phi$ and precipitation as a function of height.

## 5    Conclusion

In summary, $k$-means clustering has been used to evaluate its ability to identify different types of correlations between the vertical distributions of precipitation- and moisture-related variables. Our work shows the application and physical interpretability of using an unsaturated polytropic potential refractivity fit, $\widehat{N}$, when there is a linear temperature profile with height, which is expected in a polytropic atmosphere. Deviations from $\widehat{N}$ relate to the presence of water vapor pressure anomalies at given latitudes, longitudes, and times (Sect. 4.2). In particular, Table 1 demonstrates a visibly strong yet non-monotonic relationship between the shapes and amplitudes of $N - \widehat{N}$ vs. $e$. For instance, the moderately negative Cluster 6 for $N - \widehat{N}$ corresponds well with very moist profiles, yet the more negative Clusters $N - \widehat{N}$ correspond to only moderately moist profiles. Inversely, the mostly flat Cluster 1 for $N - \widehat{N}$ corresponds to profiles with little to no moisture (Clusters 3 and 7 for $e$) yet the most positive Cluster 5 for $N - \widehat{N}$ corresponds to profiles with low to moderate moisture. This can be explained by how the deviation of $N$ from $\widehat{N}$ will be muted if $\widehat{N}$ has been fit to a profile which is moist overall, and thereby $\left| N - \widehat{N} \right|$ will be largest when the moisture is large and relatively localized (e.g., in the presence of clouds).

$\widehat{N}$ coefficient ($\mathbf{c}$) clusters can flag physical vs. nonphysical values of observed and derived variables (Sect. 4.3, Figure 6). As shown in Figure 6, Clusters 5-7 for $\mathbf{c}$ generally correspond to temperature values that are far too low, indicating either a problem with the data from the retrievals or a profile which does not satisfy the physical assumptions made in deriving $\widehat{N}$ (see Sect. 3.1). Inversely, the values of $\mathbf{c}$ for a given profile can identify when a profile has no moisture or precipitation with very high accuracy—as shown in Figure 7, profiles with $\widehat{\Gamma} > 10^{-1}$ K/km and $\widehat{T}_0 > 280K$ have little to no precipitation. Related correlations between different $\mathbf{c}$ and $e$ clusters are also shown in Table 2, where we see that different clusters for $\mathbf{c}$ correspond to profiles with low, medium, and high water vapor pressure throughout.

Similarly, vertical distributions of $\Delta\Phi$ are found to correlate to specific vertical profiles of liquid and ice precipitation. In particular, the amplitude and shape of $\Delta\Phi$ centroids correlate with the amplitudes and shapes of LWP and IWP centroids,

respectively (Sect. 4.4, Table 3). This correlation persists across low and high levels of LWP, IWP, and both combined, thereby demonstrating a strong one-to-one relationship between $\Delta\Phi$ and water path.

In conclusion, the clustering centroids (i.e., "representative" profiles) correlate with the general magnitude of a variable for a given profile and also the general shape of that variable as a function of height. The latter is especially evident for variables that correlate with water content: $\Delta\Phi$ and the path variables. As a demonstration of how the centroids capture the magnitude of profiles in their associated clusters, consider the ice water path (IWP) clusters shown in Figure 4(c): Clusters 4 and 7 for IWP both correspond to higher-than-average ice content in their respective profiles, and a similar comparison can be drawn between Clusters 2 and 6 for IWP. Relatedly, as a demonstration of how the centroids capture the shape, consider the liquid water path (LWP) clusters shown in Figure 4(d): Clusters 2 and 5 for LWP both correspond to non-negligible water precipitation, but Cluster 5 features profiles with deeper precipitation than those in Cluster 2. Thus, clustering in the manner introduced in this study confirms its value as a tool for quality control of profiles and automates the classification of physical phenomena found across large datasets.

## Appendix A: Derivation of $\widehat{N}$

Combining the equation for hydrostatic equilibrium and the ideal gas law, we have

$$p(z) = p(z_0)\exp\left(-\frac{g}{R}\int_{z_0}^{z}\frac{ds}{T(s)}\right) \tag{A1}$$

where $g = 9.8$ g/m$^2$ is the acceleration due to gravity on Earth and $R = 287$ J·kg$^{-1}$·K$^{-1}$ is the specific gas constant for dry air. In a polytropic atmosphere, $T(z) = T(z_0) - \Gamma(z - z_0)$ for a lapse rate $\Gamma$ to be determined by a fit to the data together with $T(z_0)$. The integral in Eq. (A1) for this temperature profile can be computed as

$$-\frac{g}{R}\int_{z_0}^{z}\frac{ds}{T(s)} = \frac{g}{R\Gamma}\log\left[1 - \frac{\Gamma(z - z_0)}{T(z_0)}\right]$$

which in turn implies that (e.g., Dutton, 1976)

$$p(z) = p(z_0)\left[1 - \frac{\Gamma}{T(z_0)}(z - z_0)\right]^{\frac{g}{R\Gamma}} \tag{A2}$$

Substituting Eq. (A2) and $T(z) = T(z_0) - \Gamma(z - z_0)$ into Eq. (1) and putting a hat on $N$ since $\widehat{N}$ is the idealized model, we have

$$\widehat{N}(z) = \frac{k_1 p(z_0)}{T(z_0)\left[1 - \frac{\Gamma(z-z_0)}{T(z_0)}\right]^2} \times \left\{\left[1 - \frac{\Gamma(z - z_0)}{T(z_0)}\right]^{\frac{g}{R\Gamma}+1} + \frac{k_2 e}{k_1 p(z_0) T(z_0)}\right\} \tag{A3}$$

While $p(z_0)$ might not be available directly in a typical PRO profile (which only contains refractivity and $\Delta\Phi$), there will be data for $N(z_0)$. Hence, we solve for $p(z_0)$ in terms of $T(z_0)$ and $N(z_0)$ to constrain the number of fitting parameters.

Rewriting Eq. (1) at $z = z_0$, we have

$$k_1 p(z_0) = T(z_0) \left[ N(z_0) - \frac{k_2 e}{T(z_0)^2} \right] \tag{A4}$$

Substituting Eq. (A4) into Eq. (A3) for a constant, representative, $\widehat{e}_0$, and rewriting in terms of the fit coefficients $c_0$, $c_1$, and $c_2$, leads to Eq. (3).

## Appendix B: Numerical fitting procedure for $\widehat{N}$

Once Eq. (3) has been fit to a given profile, we can use $c_0$ to solve for $\widehat{\Gamma}$, then use this and $c_1$ to solve for $\widehat{T}_0$, and finally, use $c_2$ and $\widehat{T}_0$ to solve for the representative value of $\widehat{e}$. To do this fitting routine in practice, since $k_2, \widehat{e}, N \geq 0$, we impose the constraint $c_2 \geq 0$ and use the curve-fitting utility `optimize.least_squares` from the SciPy package (version 1.7.3) in Python 3.7.4 with the initial conditions $c_0 = 4.5$, $c_1 = 0.01 \text{ m}^{-1}$, and $c_2 = 0$ to fit Eq. (3) to each $N$ profile for all cases. For reasons which are generally internal to the default `optimize.least_squares` algorithm, the nonlinear fitting procedure either did not always converge within the preset maximum number of iterations, 10000, with prescribed error tolerances `ftol=xtol=` $10^{-12}$, or the profile in question was missing too much data for the model coefficients to be uniquely determinable—this only occurred in 5 profiles out of the 6706 in the dataset, or 0.07%. The latter could have either occurred because there were not enough data overall or because there were no refractivity data at $z_0 = 2.5$ km.

## Appendix C: Clustering algorithm for k-means

For the numerical implementation of time series $k$-means clustering, we use version 0.6.2 of the Python package `tslearn`, which provides machine learning tools for the analysis of time series data and builds on the `scikit-learn`, `scipy`, and `numpy` libraries (Tavenard et al., 2020). To run time series $k$-means clustering for all variables in the dataset, we use `tslearn.clustering.TimeSeriesKMeans` with $k = 8$ clusters, the DTW metric, a maximum of 30 iterations of the algorithm, and we fix the random state to 0 to ensure that the cluster labels stay consistent upon each run.

There are ways to estimate the most "statistically meaningful" number of clusters for a given dataset, even when not using the Euclidean metric—e.g., the average silhouette method (Rousseeuw, 1987) or the gap statistic method (Tibshirani et al., 2001)—which could give different numbers of clusters for each variable. However, to keep a consistent number of clusters for each variable, and to give some semblance of the same hierarchy in magnitude across clustering in each variable, this study uses the same number of clusters for all variables and defers to using a number that is possibly too large rather than too small.

## Appendix D: Quality-control criteria for clustering

Profiles are excluded from each cluster according to the quality-control criteria listed below.

- $\Delta\Phi$ (923 profiles excluded, or 13.76%): Files are excluded by the same criteria used for the vertical integral of $\Delta\Phi$.

- $c_0$, $c_1$, and $c_2$ (5 profiles excluded, or $0.07\%$): The fit for $\widehat{N}$ must converge, i.e., the algorithm for computing the best-fit coefficients $c_0$, $c_1$, and $c_2$ must converge, which means that there must be refractivity data at 2.5 km, there must be enough refractivity data between 2.5 km and the estimated lapse-rate tropopause (the latter of which was explained earlier), and the fit must converge within 10000 iterations for tolerance conditions `ftol=xtol=`$10^{-12}$.

- $N - \widehat{N}$ (223 profiles excluded, or $3.33\%$): Along with the same criteria related to $\widehat{N}$ used for the coefficient clusters, cases where the tropopause is below 8.2 km are skipped and three files from clustering for $N - \widehat{N}$ are taken out manually and excluded. These three files contained unphysically large values of $N$ ($N > 600$) and likely indicate an issue with retrieving the refractivity for the RO dataset.

- Water vapor pressure (33 profiles excluded, or $0.49\%$): Files are excluded by the same criteria used for the total column water vapor described in Sect. 2. It should be noted that the three files with unphysically large values of $N$ that were manually excluded from clustering for $N - \widehat{N}$ also had unphysically large water vapor pressure values. Hence, these profiles were also excluded from the clustering for water vapor pressure, thereby showing that at least some of the erroneous water vapor pressure retrievals were caused by issues with the retrieved RO refractivity.

- LWP (1 profile excluded, or $0.01\%$): Files without LWP data from 1 to 10 km are excluded.

- IWP (6 profiles excluded, or $0.09\%$): Files without IWP data from 1 to 10 km are excluded.

- TWP (6 profiles excluded, or $0.09\%$): Files without LWP or IWP data from 1 to 10 km are excluded.

*Code and data availability.* The datasets associated with this study have been uploaded to the Jet Propulsion Laboratory's GENESIS (Global Environmental & Earth Science Information System) site: https://genesis.jpl.nasa.gov/ftp/paz_pol/. Further ROHP data are available at https://paz.ice.csic.es. GPM level 1C passive MW radiometer data are openly available via the Precipitation Processing System (PPS) at NASA Goddard Space Flight Center: https://pps.gsfc.nasa.gov/.

*Author contribution.* Conceptualization: JK, MTJ, KNW; Data curation: JT, KNW, RP; Formal analysis: JK, MTJ, TK; Funding acquisition: MTJ; Investigation: All; Methodology: JK, MTJ, TK, KNW; Project administration: MTJ; Resources: MTJ, KNW; Software: All; Supervision: MTJ, TK, JT; Validation: All; Visualisation: JK, MTJ, TK; Writing – original draft preparation: JK, MTJ, TK; Writing – review & editing: JK, MTJ, TK, JT, RP

*Competing interests.* The authors declare that they have no conflict of interest.

*Acknowledgements.* This work was carried out at the Jet Propulsion Laboratory, California Institute of Technology under the JPL Visiting Student Research Program with support from NASA's NH19ZDA001N-GNSS program, under a contract with the National Aeronautics and Space Administration (80NM0018D0004), and with a stipend and teaching fellowship from the

Yale Graduate School of Arts and Sciences. The authors would like to thank Joe Turk for collecting and preparing the GPM dataset, Kuo-Nung Wang and Ramon Padullés for preparing and managing the ROHP-PAZ dataset and collocations between the GPM and ROHP-PAZ datasets, and various technical support staff at the Jet Propulsion Laboratory for their tireless help with data, equipment, and account access. We would also like to thank Chi O. Ao for helping to manage and prepare the data and software resources used in this study.

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
