# Peer review of "Cluster Analysis of Vertical Polarimetric Radio Occultation Profiles and Corresponding Liquid and Ice Water Paths From GPM Microwave Data"

_EGUsphere, 2024_

## Referee Comment (RC1)

**Manuscript Review**

**Title:** Cluster Analysis of Vertical Polarimetric Radio Occultation Profiles and Corresponding Liquid and Ice Water Paths From GPM Microwave Data

**Authors:** Jonas Katona, Manuel de la Torre Juarez, Terence L. Kubar, F. Joseph Turk, Kuo-Nung Wang, and Ramon Padulles

**Reviewer Recommendation:** Minor Revision

**Overall Quality:**

In this study, the authors evaluate the ability of k-means clustering to find relationships among polarimetric phase difference, refractivity, liquid water path (LWP), ice water path (IWP), and water vapor pressure using over two years of data matched between the GPM mission and ROHP-PAZ. They develop a refractivity model to ascertain how different types of vertical thermodynamic profiles that can occur during different precipitation scenarios are related to changes in the polytropic index and thereby vertical heat transfer rates. The authors' main conclusion is that clustering in the manner introduced in this study confirms its value as a tool for quality control of profiles and can automate the classification of physical phenomena found across large datasets, thereby avoiding the need to inspect and compare profiles individually. I believe that this work is, overall, a well-written manuscript and will provide benefits to both the RO/PRO and precipitation communities, although it will first be necessary to clarify some issues and expand on details/explanations. My recommendation for this paper is that a minor revision is necessary for publication and my review is followed.

**Specific comments:**

1. The abstract is well-written, although there should be a more definitive concluding sentence summarizing the overall main conclusion of the work (e.g., what is the usefulness or benefits of this study toward the science?)

2. Lines 39-47: There should be a brief summary of the RO 1D-Var method if the authors are going to mention how water vapor information can be extracted from refractivity profiles. A good resource to use is from Wee et al. 2022 (**https://doi.org/10.3390/rs14215614**).

3. Lines 87-88: What are the collocation criteria used in this study? These values need to be explicitly defined in this sentence (e.g., 3 hours/300 km - whatever values you used).

4. Figure 1 caption: Sampling distribution of what? This caption needs to be more detailed. I assume this is showing the locations/times of the collocations, but readers shouldn't

have to guess. Also, this also raises another question that I do not see answered in the methodology: What tangent point lat/lon is chosen for this figure? Nearest to surface? 10 km? Please add this to the paper.

5. Line 90: Why is there a data gap in Jan/Feb 2019?

6. Lines 100-101: The authors write "By checking when the retrieved temperature is above or below 273 K, we partition this integrated water content into LWP and IWP, respectively". Is it appropriate to simply use the freezing point of water to differentiate between LWP and IWP when there are often supercooled water droplets in clouds?

7. Lines 177-180: Why do the authors use this definition of the tropopause to identify its location, rather than use an established and commonly used definition, such as the lapse-rate tropopause definition from the WMO: "the lowest level at which the temperature lapse rate decreases to 2 K/km or less. To fill this condition, the average lapse rate between this specific level and all higher levels within 2 km should not exceed 2 K/km."

8. Line 197: What are some examples of "nonphysically high water vapor pressure values (> 300 hPa)" and what are some possible causes for these erroneously retrieved values?

9. Lines 229-232: The authors write a summary from previous studies: "Later studies (Muller et al., 2009; Holloway and Neelin, 2010; Emmenegger et al., 2022) demonstrate a positive relationship between precipitation and total column water vapor in the tropics, where under a certain total column water vapor value, precipitation is generally negligible in a given profile, and above a "pickup" threshold, precipitation may become non-negligible and increases exponentially." I expected to see the authors apply this same idea towards their own study, with a focus on Figure 3, especially because in lines 233-234, the authors write "we look for the precipitation pickup pattern (Holloway and Neelin, 2010) in the relationship between the total column water vapor and the total column of the PRO observable". However, I don't really see that analysis provided. As readers, yes, we can attempt to visualize a relationship in Figure 3. But I'm not sure I'm convinced by a precipitation pickup pattern in any of those panels. For example, in each of those panels, at what total column water vapor value do the authors see this "precipitation pickup pattern"? It's not easily discernable to me. Additionally, how much are those moving averages influenced by a handful of large phase difference outliers? I

think this specific analysis needs to be thought about more carefully and expanded upon in order to be considered a useful result.

10. Lines 272-273: The authors write "Furthermore, Nmodel is fit across most of the troposphere down to 2.5 km. Hence, Nmodel is most effectively sensitive to concentrated moisture anomalies within narrow bands of the troposphere." I'm struggling to understand this statement. Wouldn't it be the opposite (e.g., the Nmodel would be least sensitive to concentrated moisture anomalies since it is fit across most of the troposphere, therefore missing those thin moisture anomaly layers)? The modeled N should not be as effective in regions where large and sharp moisture anomalies/gradients are present, such as in deep convection, where rapid vertical moisture transport commonly occurs. Can the authors elaborate on what their statement means (both in response to this question as well as in the paper)?

11. Line 279: The authors write "Fig. 4 shows three examples where $N - N_{model}$ does not correlate strongly with $\Delta\Phi$." However, it is not clear to me how Fig. 4a follows that statement – how would the authors expect those refractivity profiles to look if they were correlated with the phase difference profile?

12. Lines 291-292: Do the authors have any suggestions as to what the possible retrieval issues might be?

**Technical Corrections:**

1. Lines 32-38: You don't need to start new paragraphs so often, as it is unnecessary to have a separate paragraph for a single sentence.

2. Lines 36-38: This has also been done in recent years using both TRMM and GPM for both tropical and midlatitude deep convection – add Johnston et al. 2018 (**https://doi.org/10.1002/2017JD027120**) and Johnston et al. 2022 (**https://doi.org/10.3390/atmos13020196**).

3. Line 51, along with many other locations: The authors seem to be differentiating between the liquid and ice phases of water by consistently writing "*precipitation or ice*". Precipitation generally encompasses any form of water that forms and falls to the earth, regardless of whether it is ice or liquid. Thus, I recommend changing any of these instances in the paper to "liquid or ice" or something along these lines. Or are the authors

trying to differentiate between precipitating and non-precipitating features? Because non-precipitating features can also be either liquid or ice.

4. Lines 74-75: delete "from PRO derived refractivity and $\Delta\Phi$, to model-inferred water vapor, water path, and ice path".

5. Line 260: change "relative humidity" to "specific humidity". You can still have a higher relative humidity in regions where there is very little moisture present, and as a result, you would not see a big impact to a refractivity profile.

---

## Author Comment (AC1)

**Responses to Referee 1**

**Summary:** The authors greatly appreciate the referee for their constructive feedback and suggestions. The referee has raised several concerns to be addressed, including a detailed description of the collocation criteria, clarification of the retrieval method used to derive variables from RO refractivity, a discussion on the sources for retrieval errors, and stronger justifications for claims related to the potential refractivity fit, $\widehat{N}$. The referee also raised important questions on the significance and robustness of the results presented in Section 4.1. In response, the authors have substantially revised the manuscript to address these concerns.

A one-by-one response to the referee's broader concerns can be found below. For brevity, we refer to sections as S (Section 4 is S4), and to lines as "L" (line XXX is LXXX). Each specific comment from the referee has been included in italics and then we specify how it was addressed in detail under "Response." Unless otherwise noted, the line numbers in this response correspond to those in the revised version with the changes marked.

**Responses to the referee's specific comments**

1. *The abstract is well-written, although there should be a more definitive concluding sentence summarizing the overall main conclusion of the work (e.g., what is the usefulness or benefits of this study toward the science?)*

   **Response:** L2-8 add three sentences to summarize the capabilities of PRO and the motivation for this work. Furthermore, L23-32 have been added at the beginning of the introduction to describe the conclusions of a study combining models and data explaining the importance of getting vertical distributions of temperature and moisture associated to precipitation.

2. *Lines 39-47: There should be a brief summary of the RO 1D-Var method if the authors are going to mention how water vapor information can be extracted from refractivity profiles. A good resource to use is from Wee et al. 2022.*

   **Response:** L64-68 have added a sentence in the former lines 39–47 mentioning the RO 1D-Var method as a common approach for extracting water vapor from RO refractivity, and have included the suggested reference to Wee et al. 2022. However, the dataset used in this study relies on JPL retrievals, which apply the direct method in Hajj et al. 2002 instead of 1D-Var.

3. *Lines 87-88: What are the collocation criteria used in this study? These values need to be explicitly defined in this sentence (e.g., 3 hours/300 km - whatever values you used).*

   **Response:** L136-139 now after the former lines 87-88 describe the collocation criteria, *that the GPM passive microwave (PMW) satellite overpass had to occur within $\pm15$ minutes of the ROHP-PAZ observation, and the ROHP-PAZ observation location had to fall within the PMW satellite's swath,* along with a source where the reader can find further details.

4. *Figure 1 caption: Sampling distribution of what? This caption needs to be more detailed. I assume this is showing the locations/times of the collocations, but readers shouldn't have to guess. Also, this also raises another question that I do not see answered in the methodology: What tangent point lat/lon is chosen for this figure? Nearest to surface? 10 km? Please add this to the paper.*

   **Response:** Figure 1 caption now describes that it shows the sampling distributions for the collocations between the GPM and ROHP-PAZ datasets. Furthermore, L136-139 now state

that for each ROHP-PAZ observation, the tangent point chosen was that of the lowest RO ray.

5. *Line 90: Why is there a data gap in Jan/Feb 2019?*

   **Response:** We added a footnote to L141 explaining that technical issues with the processing of ROHP-PAZ retrievals from January and February 2019 prevented the creation of the collocated dataset between GPM and ROHP-PAZ for these two months. Although the ROHP-PAZ retrievals from this period have since been corrected and are now available in the latest ROHP-PAZ dataset, the analysis in this study was completed before updated collocations could be made. It does not change the conclusions of our study.

6. *Lines 100-101: The authors write "By checking when the retrieved temperature is above or below 273 K, we partition this integrated water content into LWP and IWP, respectively." Is it appropriate to simply use the freezing point of water to differentiate between LWP and IWP when there are often supercooled water droplets in clouds?*

   **Response:** L153-158 now acknowledge that using only the freezing point to distinguish between LWP and IWP is a simplification, as noted in the former lines 100–101. However, this study does not rely on the assumption that IWP consists exclusively of ice to draw any critical conclusions. The primary analyses do not reveal significant differences in the correlations between LWP and IWP (as defined in this dataset) with other variables, except for the observation that IWP profiles tend to exhibit deeper peaks than those of LWP (see Figure 5).

   For comparison, any distinction between water and ice in GPM products is done via consultation with temperature information from nearby (in time and space) numerical weather predictions or reanalysis [6]. The GPM radiometers do not directly measure temperature, which limits the ability of GPM data to determine phase uncertainty between liquid and ice without relying on model data, which lacks the certainty of separating supercooled liquid from ice water.

   Given the much stronger PRO phase delay response to ice than to water [5], the authors believe that a careful analysis of the shape of the PRO phase delay to ice could enable the detection of supercooled water in the future. Ice would be expected to increase the polarimetric phase difference with height as the signal propagates above the clouds and precipitation region, whereas non-precipitating supercooled water would induce little to no polarimetric phase difference [5, 2]. This possibility opens an avenue for future research.

7. *Lines 177-180: Why do the authors use this definition of the tropopause to identify its location, rather than use an established and commonly used definition, such as the lapse rate tropopause definition from the WMO: "the lowest level at which the temperature lapse rate decreases to 2 K/km or less. To fill this condition, the average lapse rate between this specific level and all higher levels within 2 km should not exceed 2 K/km."*

   **Response:** L238-243 have been reworded. The authors expect this height to be often close to the lapse rate tropopause, they understand that this is not a conventional measure of the tropopause, but rather, a lower bound on where the linear temperature (i.e., constant lapse rate) profile assumption would generally start to break down occasionally because of clouds or gravity waves related to convection, but is high enough to have a temperature close to that of the tropopause.

8. *Line 197: What are some examples of "'nonphysically high water vapor pressure values (>300 hPa)" and what are some possible causes for these erroneously retrieved values?*

12. *Lines 291-292: Do the authors have any suggestions as to what the possible retrieval issues might be?*

**Response to both (8) and (12):** To the first question, the authors have added in L169-174 a new description of the technique used to identify profiles with unphysically large water vapor pressure profiles in Section 2.

It was misleading to emphasize 300 hPa as an a priori threshold in the original text instead of merely stating that 300 hPa (which in fact could have been set to 250 hPa) was an *observed* lower bound for the maximum water vapor pressure in anomalous profiles. This inaccurate wording has been removed.

As described in the first paragraph of Section 2 and in response to comment 2 above, the water vapor pressure in this study is derived from RO refractivity using the direct method [1], with the background NCEP temperature assumed accurate. Keeping these unphysical values help identify profiles in which there was an inconsistency between the model and RO observed refractivities. Negative or otherwise unrealistic values would serve as a quality control flag. One result was to discover that cluster analysis could be used to identify and thereby exclude invalid retrievals.

9. *Lines 229-232: The authors write a summary from previous studies: "Later studies (Muller et al., 2009; Holloway and Neelin, 2010; Emmenegger et al., 2022) demonstrate a positive relationship between precipitation and total column water vapor in the tropics, where under a certain total column water vapor value, precipitation is generally negligible in a given profile, and above a "pickup" threshold, precipitation may become non-negligible and increases exponentially." I expected to see the authors apply this same idea towards their own study, with a focus on Figure 3, especially because in lines 233- 234, the authors write "we look for the precipitation pickup pattern (Holloway and Neelin, 2010) in the relationship between the total column water vapor and the total column of the PRO observable." However, I don't really see that analysis provided. As readers, yes, we can attempt to visualize a relationship in Figure 3. But I'm not sure I'm convinced by a precipitation pickup pattern in any of those panels. For example, in each of those panels, at what total column water vapor value do the authors see this "precipitation pickup pattern?" It's not easily discernable to me. Additionally, how much are those moving averages influenced by a handful of large phase difference outliers? I think this specific analysis needs to be thought about more carefully and expanded upon in order to be considered a useful result.*

    **Response:** The authors have now reframed much of Section 4.1 to test primarily how well $\Delta\Phi$ correlates positively with the total column water vapor. The tables in this section capture a somewhat significant correlation on average but quite poor for individual profiles. Given the sparsity of our dataset, particularly in the tropics, we echo the reviewer's remark that we cannot state anything conclusive on how well polarimetry can recover the precipitation pickup pattern when using $\Delta\Phi$ as a proxy for precipitation. This might give more relevance to the analyses present in the rest of Section 4 which instead relate $N - \widehat{N}$ to the water vapor pressure across individual profiles. All of this has now been stated in the revised text for S4.1.

10. *Lines 272-273: The authors write "Furthermore, Nmodel is fit across most of the troposphere down to 2.5 km. Hence, Nmodel is most effectively sensitive to concentrated moisture anomalies within narrow bands of the troposphere." I'm struggling to understand this statement. Wouldn't it be the opposite (e.g., the Nmodel would be least sensitive to concentrated moisture anomalies since it is fit across most of the troposphere, therefore missing those thin moisture anomaly layers)? The modeled N should not be as effective in regions where large and sharp moisture anomalies/gradients are present, such as in deep convection, where rapid vertical moisture transport commonly occurs. Can the authors elaborate on what their statement means (both in response to this question as well as in the paper)?*

    **Response:** The original wording may have been ambiguous. The referee is correct that $\widehat{N}$ tends to smooth out and neglect sharp, concentrated moisture anomalies because it is fit across broad portions of the troposphere.

Thanks, the authors' original intent was to state that the *differences between $\widehat{N}$ and the measured refractivity N* would be most prominent in regions with sharp moisture gradients. This is advanced in L221-222. The authors agree that using "sensitive to concentrated moisture anomalies" to describe $\widehat{N}$ itself was a misleading choice of wording. We think that now L349-352 and L376-383 better reflect the intended meaning.

11. *Line 279: The authors write "Fig. 4 shows three examples where N-Nmodel does not correlate strongly with $\Delta\Phi$." However, it is not clear to me how Fig. 4a follows that statement – how would the authors expect those refractivity profiles to look if they were correlated with the phase difference profile?*

    **Response:** L346-352 better clarify the intended description of Fig. 5(a). Note that while there is a hump in water vapor pressure from roughly 2.5 km to 4.5 km, $\Delta\Phi$ fluctuates without a clear trend and with the same small amplitude throughout the entire profile. If $e$ and $\Delta\Phi$ were correlated in this case, the authors would expect $\Delta\Phi$ to similarly show a hump or at least an average increase in amplitude near the bottom of the profile. Meanwhile, to the point of this section, $N$ and $\widehat{N}$ also show a slight deviation from each other from roughly 2.5 km to 4.5 km. Likewise, in Fig. 5(b), the water vapor pressure roughly decreases exponentially as height increases; the deviation of $\widehat{N}$ from $N$ is quite notable throughout the profile until the water vapor pressure is near zero. However, $\Delta\Phi$ remains near zero or slightly negative throughout the entire profile.

    Finally, to address the other referee's concerns about the introduction of the figures in the main text, the authors have moved the final paragraph of Section 4.2 to the beginning of Section 4.2 and reworded much of it.

**Technical corrections**

1. *Lines 32-38: You don't need to start new paragraphs so often, as it is unnecessary to have a separate paragraph for a single sentence.*

    **Response:** The authors have combined all single-sentence paragraphs with either their preceding or proceeding paragraphs.

2. *Lines 36-38: This has also been done in recent years using both TRMM and GPM for both tropical and midlatitude deep convection – add Johnston et al. 2018 and Johnston et al. 2022.*

    **Response:** [3, 4] are now cited at the end of the former line 35, and we also added a sentence in lines L55-58 describing the relevance of [3, 4].

3. *Line 51, along with many other locations: The authors seem to be differentiating between the liquid and ice phases of water by consistently writing "precipitation or ice." Precipitation generally encompasses any form of water that forms and falls to the earth, regardless of whether it is ice or liquid. Thus, I recommend changing any of these instances in the paper to "liquid or ice" or something along these lines. Or are the authors trying to differentiate between precipitating and non-precipitating features? Because non precipitating features can also be either liquid or ice.*

    **Response:** We have changed all such cases to something more specific, depending on the context.

4. *Lines 74-75: delete "from PRO derived refractivity and $\Delta\Phi$, to model-inferred water vapor, water path, and ice path."*

    **Response:** We have removed the indicated phrase from the manuscript.

5. *Line 260: change "relative humidity" to "specific humidity." You can still have a higher relative humidity in regions where there is very little moisture present, and as a result, you would not see a big impact to a refractivity profile.*

**Response:** Thank you; we have followed through with the referee's suggestion.

**References**

[1]  G. Hajj et al. "A technical description of atmospheric sounding by GPS occultation". In: *Journal of Atmospheric and Solar-Terrestrial Physics* 64.4 (2002), pp. 451–469. ISSN: 1364-6826. DOI: `10.1016/S1364-6826(01)00114-6`.

[2]  D. Hotta, K. Lonitz, and S. Healy. "Forward operator for polarimetric radio occultation measurements". In: *Atmospheric Measurement Techniques* 17.3 (2024), pp. 1075–1089. DOI: `10.5194/amt-17-1075-2024`.

[3]  B. R. Johnston, F. Xie, and C. Liu. "Relationships between Extratropical Precipitation Systems and UTLS Temperatures and Tropopause Height from GPM and GPS-RO". In: *Atmosphere* 13.2 (2022). ISSN: 2073-4433. DOI: `10.3390/atmos13020196`.

[4]  B. R. Johnston, F. Xie, and C. Liu. "The Effects of Deep Convection on Regional Temperature Structure in the Tropical Upper Troposphere and Lower Stratosphere". In: *Journal of Geophysical Research: Atmospheres* 123.3 (2018), pp. 1585–1603. DOI: `10.1002/2017JD027120`.

[5]  R. Padullés, E. Cardellach, and F. J. Turk. "On the global relationship between polarimetric radio occultation differential phase shift and ice water content". In: *Atmospheric Chemistry and Physics* 23.3 (2023), pp. 2199–2214. DOI: `10.5194/acp-23-2199-2023`.

[6]  F. J. Turk et al. ": Interpretation of the Precipitation Structure Contained in Polarimetric Radio Occultation Profiles Using Passive Microwave Satellite Observations". In: *Journal of Atmospheric and Oceanic Technology* 38.10 (2021), pp. 1727–1745. DOI: `10.1175/JTECH-D-21-0044.1`.

---

## Author Comment (AC2)

**Responses to Referee 2**

**Summary:** The authors thank the referee for their constructive feedback and suggestions. We have made extensive revisions to better contextualize the scientific value and relevance of this work—both in the analyses and in information on the retrievals— with other studies that use RO or PRO data.

A summary of major changes and an overview of how these address the referee's broader concerns can be found below. For brevity, we refer to sections as S (Section 4 is S4), and to lines as "L" (line XXX is LXXX). Each specific comment from the referee has been included in italics and then we specify how it was addressed in detail under "Responses" Unless otherwise noted, the line numbers in this response correspond to those in the revised version with the changes marked.

**Responses to referee comments**

1. *Line 27: better to have units for $k_1$ and $k_2$.*

   **Response:** $k_1$ and $k_2$ are empirically determined proportionality constants that are provided without units in the literature (e.g., [4, 7, 8]), even it makes sense that $k_1$ and $k_2$ differ by a ratio of temperature units in K as we now mention in L43-44.

2. *Line 40: 1DVAR (e.g., UCAR/COSMIC) should be briefed here when talking about "wet" profiles, i.e., temperature and humidity.*

   **Response:** The dataset used in this study relies on JPL retrievals, which apply the direct method [2] rather than the 1D-Var method. However, we also added a reference to the 1D-Var method in L64-68.

3. *Line 53, I think there is another important paper by Padullés et al. (2022) in which they discussed the sensitivity of PRO simulation to frozen hydrometeors based. Could the authors provide any insights into the separation of frozen hydrometeors rather than the two water phases (ice and precipitation) used? Perhaps a few sentences could be added to the discussion or introduction.*

   *Additionally, what are the authors' thoughts on how their results can be generalized to high-resolution NWP model states? Specifically, could the authors offer perspectives on the potential application of their methods using NWP model data, rather than the GPM data used in this study?*

   **Response:** The analysis in [5] focused on the RO rays above the freezing temperature level. It remains hard to untangle the structure of water phases along the RO ray, since $\Delta\Phi$ is an integrated value along a ray path that can cross both liquid and ice phases. Doing so is beyond the scope of this paper. It remains an open question how or even if it is possible to differentiate between ice and liquid water precipitation using PRO data alone.

   L89-95 in the introduction discuss how PRO data can be used to distinguish between ice and liquid precipitation, as well as precipitating vs. non-precipitating features, and how some questions are still open.

   The authors do not feel equipped to answer the second question. L69-71 briefly refer to PRO being most valuable where they disagree with NWP models—that is why this study did not use NWP data—but a follow-on effort using NWP instead of GPM and limited to the cases where PRO and NWP refractivities agree, would have increased statistics. In parallel, this combination of NWP and PRO could hold relevance for assimilating PRO data

into NWP frameworks [3, 6, 10]. This remains a work in progress, and several preliminary studies on PRO data assimilation in NWP were presented at the 2nd PAZ-Polarimetric Radio Occultations User Workshop and summarized in [9].

4. *Line 60: "Part of the challenge is that a given $\Delta\Phi$ at a specific height may be caused by both ice or precipitation." Any other potential causes? Could the authors provide some discussion about the challenges in representing $\Delta\Phi$ by model states?*

    **Response:** The wording of "ice or precipitation" has been changed to refer to the separation of ice and liquid water precipitation. To answer the first question, L89-95 have also added some words on the documented influence of anisotropic ice crystals on PRO [5] and the negligible influence of smaller particles like aerosols and non-precipitating cloud droplets on GNSS PRO [3].

5. *Line 66: The k-means cluster analysis is present without introduction. Could the authors provide a brief introduction. What are the major advantages using this analysis? Why do the authors use this method? Any references?*

    **Response:** L97-101 now have added an introductory paragraph before the former line 66. This paragraph briefly defines cluster analysis and cites several papers, a survey, and a textbook that demonstrate the use of $k$-means clustering in analyzing climate and atmospheric data.

6. *Line 90: It is better to provide a brief introduction about the GPM data. For instance, whether it is gridded data. What are the spatial and temporal resolution? How the matching is being done?*

    **Response:** L121-124 have added a description of the spatial and temporal resolution of the Level 2 GPM data. A sentence has been added in L136-139, after former lines 87-88, to describe the collocation criteria used for matching:

7. *Line 95 and Fig. 1b: What caused the missing period in Jan/Feb 2019? If the monthly coverage is not equal and the seasonality is not studied in this paper, Fig. 1b does not seem useful and can be removed.*

    **Response:** We added a footnote to L141 explaining that technical issues with the processing of ROHP-PAZ retrievals from January and February 2019 prevented the creation of the collocated dataset between GPM and ROHP-PAZ for these two months. Although the ROHP-PAZ retrievals from this period have since been corrected and are now available in the latest ROHP-PAZ dataset, the analysis in this study was completed before updated collocations could be made. It does not change the conclusions of our study.

    However, we included Figure 1(b) to provide comprehensive documentation of the dataset, including the seasonal distribution of the coincidences (e.g., a seasonal bias). Since the main text includes a plot of the latitudinal distribution of the dataset (Figure 1(a)), both convey the spatiotemporal distribution of the dataset.

8. *Line 197: Do the authors really mean 300hpa? 300hpa of water vapor pressure is way beyond the quality. How does this the QC threshold of "nonphysically high water vapor pressure values" come from?*

    **Response:** To the first question, the authors have added in L169-174 a new description of the technique used to identify profiles with unphysically large water vapor pressure profiles in Section 2. It was misleading to emphasize 300 hPa as an a priori threshold in the original text instead of merely stating that 300 hPa, which could have been set to 250 hPa, was an *observed* lower bound for the maximum water vapor pressure in anomalous profiles. This inaccurate wording has been removed.

    As described in the response to comment 2 above, the water vapor pressure in this study is derived from RO refractivity using the direct method [2]. This occasionally leads to unphysical

negative values that help identify profiles in which there was an inconsistency between the model and RO observed refractivity. Negative or otherwise unrealistic values serve as a quality control flag. One result of the cluster study was to discover that cluster analysis separated a group associated with unphysical retrievals.

9. *Line 228: "Bretherton et al. (2004) showed a relationship between precipitation and total column water vapor over the tropics." What did this paper say? Could the authors add the main findings of such "relationship?"*

10. *Line 231: Could the authors add a bit more details about the "pickup?" was there any particular pickup values discussed in these papers?*

    **Response (to both (9) and (10)):** L304-312 have been expanded to summarize the relevant result from [1], an exponentially increasing relationship between precipitation and total column relative humidity over the tropics. [1] does not directly prescribe a "start" to the relationship they identify.

11. *Line 235 and Fig. 3: 1) Besides caption, a brief description should be given for each figure in the content before they are being discussed. 2) if the authors really think they want to talk about Panel d first, they may want to re-arrange the panels so they can discuss by alphabetic orders in the content. 3) As the authors state that "we look for the precipitation pickup pattern." The pattern is not actually being discussed.*

    **Response:**

    (a) We have added a brief introduction and description of all panels for Fig. 3 in the main body of the text.

    (b) In the revised manuscript, the authors no longer talk about Fig. 3(d) first, but Fig. 3 as a whole.

    (c) We have drastically restructured and changed the conclusions of Section 4.1; in the revised manuscript, the precipitation pickup pattern is no longer the focus of Fig. 3, but rather a more general, positive relationship between accumulated $\Delta\Phi$ and total column water vapor. The statistics in the tables 4 show some correlation in the mean values, but the figures are insufficient to convincingly claim that a precipitation pickup pattern is clearly discernible in Fig. 3.

12. *Line 238: "There is also an apparent total column water vapor threshold after which $\Delta\Phi$, the PRO signature of precipitation, starts increasing at a faster rate..." I suggest the authors merge/reorganize this sentence with the previous paragraph where they discussed the relationship in literatures between total column water vapor threshold and $\Delta\Phi$.*

    **Response:** In rewritting Section 4.1, we deleted that sentence (former line 238).

13. *Line 279: Again, this paragraph describing what Fig. 4 presents should have appeared much earlier.*

    **Response:** We have moved the paragraph describing Fig. 4 from the end of S4.2 to the beginning of S4.2, just before Fig. 4 is first referenced and interpreted.

14. *Line 57: what does "CloudSat" mean particularly here? Any description?*

    **Response:** L83-85 briefly introduce CloudSat.

15. *Line 79: model (3)?*

    **Response:** The phrase "model (3)" within the former line 79 has been rewritten as, "potential refractivity model."

**References**

[1] C. S. Bretherton, M. E. Peters, and L. E. Back. "Relationships between Water Vapor Path and Precipitation over the Tropical Oceans". In: Journal of Climate 17.7 (Apr. 2004), pp. 1517–1528. DOI: 10.1175/1520-0442(2004)017<1517:rbwvpa>2.0.co;2.

[2] G. Hajj et al. "A technical description of atmospheric sounding by GPS occultation". In: Journal of Atmospheric and Solar-Terrestrial Physics 64.4 (2002), pp. 451–469. ISSN: 1364-6826. DOI: 10.1016/S1364-6826(01)00114-6.

[3] D. Hotta, K. Lonitz, and S. Healy. "Forward operator for polarimetric radio occultation measurements". In: Atmospheric Measurement Techniques 17.3 (2024), pp. 1075–1089. DOI: 10.5194/amt-17-1075-2024.

[4] A. Kliore et al. "Preliminary Results on the Atmospheres of Io and Jupiter from the Pioneer 10 S-Band Occultation Experiment". In: Science 183.4122 (Jan. 1974), pp. 323–324. DOI: 10.1126/science.183.4122.323.

[5] R. Padullés, E. Cardellach, and F. J. Turk. "On the global relationship between polarimetric radio occultation differential phase shift and ice water content". In: Atmospheric Chemistry and Physics 23.3 (2023), pp. 2199–2214. DOI: 10.5194/acp-23-2199-2023.

[6] B. Ruston and S. Healy. "Forecast Impact of FORMOSAT-7/COSMIC-2 GNSS Radio Occultation Measurements". In: Atmospheric Science Letters 22.3 (2021), e1019. DOI: https://doi.org/10.1002/asl.1019.

[7] E. K. Smith and S. Weintraub. "The Constants in the Equation for Atmospheric Refractive Index at Radio Frequencies". In: Proceedings of the IRE 41.8 (1953), pp. 1035–1037. DOI: 10.1109/JRPROC.1953.274297.

[8] M. de la Torre Juárez et al. "Signatures of Heavy Precipitation on the Thermodynamics of Clouds Seen From Satellite: Changes Observed in Temperature Lapse Rates and Missed by Weather Analyses". In: Journal of Geophysical Research: Atmospheres 123.23 (2018), pp. 13,033–13, 045. DOI: 10.1029/2017JD028170.

[9] F. J. Turk et al. "Advances in the Use of Global Navigation Satellite System Polarimetric Radio Occultation Measurements for NWP and Weather Applications". In: Bulletin of the American Meteorol 105.6 (2024), E905–E914. DOI: 10.1175/BAMS-D-24-0050.1.

[10] K.-N. Wang et al. "The Effects of Heavy Precipitation on Polarimetric Radio Occultation (PRO) Bending Angle Observations". In: Journal of Atmospheric and Oceanic Technology 39.2 (Feb. 2022), pp. 149–161. DOI: 10.1175/jtech-d-21-0032.1.